# A genome-wide CRISPR/Cas9 screen identifies calreticulin as a selective repressor of ATF6α

**Joanne Tung, Lei Huang, Ginto George, Heather P Harding, David Ron, Adriana Ordonez***

Cambridge Institute for Medical Research (CIMR), University of Cambridge, Cambridge Biomedical Campus, Cambridge, United Kingdom

**Abstract** Activating transcription factor 6 (ATF6) is one of three endoplasmic reticulum (ER) transmembrane stress sensors that mediate the unfolded protein response (UPR). Despite its crucial role in long-term ER stress adaptation, regulation of ATF6 alpha (α) signalling remains poorly understood, possibly because its activation involves ER-to-Golgi and nuclear trafficking. Here, we generated an ATF6α/Inositol-requiring kinase 1 (IRE1) dual UPR reporter CHO-K1 cell line and performed an unbiased genome-wide CRISPR/Cas9 mutagenesis screen to systematically profile genetic factors that specifically contribute to ATF6α signalling in the presence and absence of ER stress. The screen identified both anticipated and new candidate genes that regulate ATF6α activation. Among these, calreticulin (CRT), a key ER luminal chaperone, selectively repressed ATF6α signalling: Cells lacking CRT constitutively activated a BiP::sfGFP ATF6α-dependent reporter, had higher BiP levels and an increased rate of trafficking and processing of ATF6α. Purified CRT interacted with the luminal domain of ATF6α *in vitro* and the two proteins co-immunoprecipitated from cell lysates. CRT depletion exposed a negative feedback loop implicating ATF6α in repressing IRE1 activity basally and overexpression of CRT reversed this repression. Our findings indicate that CRT, beyond its known role as a chaperone, also serves as an ER repressor of ATF6α to selectively regulate one arm of the UPR.

***For correspondence:**
aog23@cam.ac.uk

## eLife assessment

In this **important** study, the authors explore ER stress signaling mediated by ATF6 using a genome-wide gene depletion screen. They find that the ER chaperone Calreticulin binds and directly represses ATF6, a new and intriguing function for Calreticulin. The evidence presented is **convincing**, based on CHO genetics and biochemical analysis.

## Introduction

The endoplasmic reticulum (ER), and its associated chaperones, constitutes the major cellular compartment for the synthesis, folding, and quality control of secretory proteins (*Sun and Brodsky, 2019*). An imbalance between synthesis and folding can lead to ER stress, potentially resulting in ER dysfunction and pathological conditions (*Wang and Kaufman, 2016*). To restore cellular homeostasis, adaptive pathways, collectively known as the unfolded protein response (UPR), are activated. The UPR is coordinated by three known ER stress transducers: inositol-requiring kinase 1 (IRE1), protein kinase R-like endoplasmic reticulum kinase (PERK), and activating transcription factor 6 (ATF6) (*Walter and Ron, 2011*). Of the three arms of the UPR, the mechanisms that govern ER stress-dependent ATF6 activation remain the least well understood, even though ATF6 mediates much of the ER-stress-induced

changes in gene expression observed in vertebrates. Thus, ATF6 is specialised in the regulation of quality control proteins in the ER (*Adachi et al., 2008*) and complete loss of ATF6 impairs survival upon ER stress in cellular and animal models (*Wu et al., 2007*).

ATF6 is a type II transmembrane ER glycoprotein containing an N-terminal portion comprising its basic leucine zipper (b-ZIP), DNA-binding and transactivation domains and a C-terminal ER luminal domain (LD) that senses stress and regulates ATF6α activity . Vertebrates have two isoforms: ATF6α and ATF6β. The ATF6α isoform dominates the regulation of UPR target genes in mammalian cells and is the focus of this study. Unlike IRE1 and PERK, which remain in the ER after UPR activation, ATF6 translocates from the ER to the Golgi, where it is sequentially cleaved by two serine protease, site-1 (S1P) and site-2 (S2P) proteases (*Haze et al., 1999*) to release its soluble active N-cytosolic domain (N-ATF6α) that is further translocated into the nucleus. There, N-ATF6α acts as a potent but short-live transcription factor (*George et al., 2020*) that binds ER stress response elements (ERSE-I and -II) in the promoter regions of UPR target genes in complex with the general transcription factor NF-Y and YY1 to upregulate the expression of a large number of ER chaperones (*Li et al., 2000*; *Yoshida et al., 2000*). Among those, the gene encoding the ER Hsp70 chaperone BiP (*GRP78*) has been well documented as an important ATF6α target (*Adachi et al., 2008*).

At ER level, it has been proposed that ATF6α is retained by its association with BiP (*Shen et al., 2002*), yet the existence of additional factors influencing its ER localisation remains uncertain. Redox regulation has also been proposed to keep ATF6α in the ER as a mixture of monomeric and multimeric disulphide-linked forms (*Nadanaka et al., 2007*) that upon ER stress are reduced by protein disulphide isomerase (PDI) family members (such as ERp18 or PDIA5), promoting the transit of a reduced monomeric ATF6α population (*Higa et al., 2014*; *Oka et al., 2019*). The Coat Protein Complex II (COPII)-coated vesicles have been implicated in the trafficking of ATF6α (*Schindler and Schekman, 2009*; *Lynch et al., 2012*) and the ATF6α_LD's *N*-glycosylation state has also been proposed to influence its ER trafficking (*Hong et al., 2004*), yet the details of these processed remain incompletely understood. Furthermore, it is widely recognised that a key step of ATF6α activation occurs in the Golgi, the site of proteolytic processing. This is reflected in the inability of cells lacking S2P to trigger BiP transcription in response to ER stress (*Ye et al., 2000*). Hence, although there is a relatively clear understanding of the events triggering ATF6α cleavage and activation upon reaching the Golgi, much remains unknown about the factors influencing ATF6α regulation in the ER.

ATF6α signalling also involves crosstalk with the IRE1 pathway. This is partly mediated by direct heterodimerisation of N-ATF6α with XBP1s, the active, spliced form (*Yamamoto et al., 2007*; *Yamamoto et al., 2004*). However, the relationship between the ATF6α and IRE1 pathway is complex: (1) XBP1 and IRE1 genes are transcriptional targets of activated ATF6α (*Yoshida et al., 2001*), (2) XBP1s and IRE1 have some different target genes (*Shoulders et al., 2013*), and (3) ATF6α activity can also repress IRE1 signalling (*Walter et al., 2018*).

Here, we established an ATF6α/IRE1 dual UPR reporter cellular model to conduct an unbiased genome-wide CRISPR/Cas9 screen focussed on ATF6α. Complemented by bioinformatic analysis, post-screen hit validation using targeted gene editing and *in vitro* biophysical assays we report on a comprehended analysis of a complex signalling pathway.

## Results

### A dual UPR reporter cell line to monitor ATF6α and IRE1 activity

In vertebrates, the transcriptional activation of BiP is predominantly governed by ATF6α (*Adachi et al., 2008*; *Adamson et al., 2016*). To dynamically monitor ATF6α activity in live cells, a 'landing pad' cassette, containing recombination sites and expressing the *Cricetulus griseus* (*cg*) BiP promoter region fused with a super folder green fluorescent protein (BiP::sfGFP), was targeted to the *ROSA26* safe-harbour locus of Chinese hamster ovary cells (CHO-K1) using CRISPR/Cas9 (*Gaidukov et al., 2018*; *Figure 1—figure supplement 1A*) in a previously characterised cell line (*CHO-XC45*) in the lab carrying an integrated XBP1s::mCherry reporter transgene monitoring the IRE1 pathway (*Harding et al., 2019*). Flow cytometry analysis of these ATF6α/IRE1 dual UPR reporter cells (referred as *XC45-6S*) treated with ER stress-inducing agents such as tunicamycin (which inhibits *N*-glycosylation), 2-deoxy-D-glucose (which inhibits glycolysis), or thapsigargin (which depletes ER calcium), revealed time-dependent activation of both UPR arms with a broad dynamic range (*Figure 1A*, top panel, and

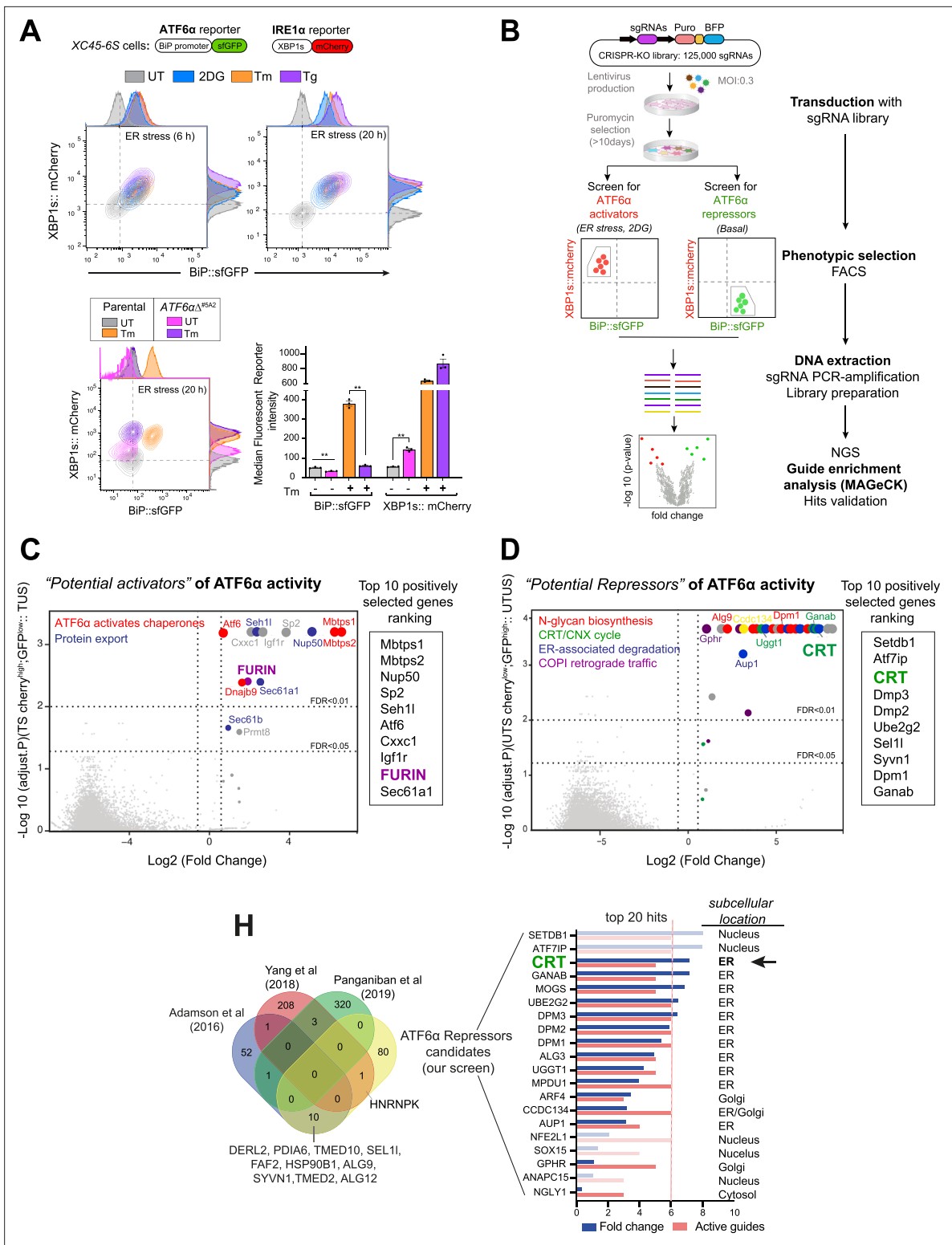

**Figure 1.** High-throughput CRISPR/Cas9 screens to identify new modulators of ATF6α signalling. (**A**) Characterisation of the *XC45-6S* cells, a dual unfolded protein response (UPR) reporter CHO-K1 cell line stably expressing XBP1s::mCherry to report on IRE1 activity and BiP::sfGFP to report on ATF6α activity was generated to perform the CRISPR/Cas9 screens. *Top panel*: Two-dimensional contour plots of BiP::sfGFP and XBP1s::mCherry signals in untreated cells (UT, grey) and cells treated with the endoplasmic reticulum (ER) stressors 2-deoxy-D-glucose (2DG, 4 mM, blue), tunicamycin (Tm, 2.5 μg/ml, orange), or thapsigargin (Tg, 0.5 μM, purple) over a short (6 hr) and long period of time (20 hr). Histograms of the signal in each channel are displayed on the oppsite axes. *Lower panel*: Contour plots as in 'A, top panel' from *XC45-6S* cells (parental) and a knockout *ATF6αΔ* clone (*ATF6αΔ#5A2*)

*Figure 1 continued on next page*

*Figure 1 continued*

generated by CRISPR/Cas9 gene editing. Cells were analysed under basal (UT) and ER stress conditions (Tm, 2.5 µg/ml, 20 hr). The bar graph shows the mean ± SEM of the median fluorescent reporter intensity normalised to untreated cells from three independent experiments. Statistical analysis was performed by a two-sided unpaired Welch's *t*-test and significance is indicated by asterisks (**p < 0.01). (**B**) Flowchart illustrating the two parallel CRISPR/Cas9 screenings aimed to identify genetic ATF6α regulators. ATF6α/IRE1 dual reporter cells were transduced with a CHO sgRNA CRISPR knockout (KO) library. After puromycin selection, the transduced population was split into two subpopulations, subjected to either ER stress with 2DG treatment (4 mM) for 18 hr or left untreated (basal) to identify the activators and repressors of ATF6α, respectively. A BiP::sfGFP$^{low}$;XBP1s::mCherry$^{high}$ population under ER stress and a BiP::sfGFP$^{high}$; XBP1s::mCherry$^{low}$ population devoid of ER stress were collected through fluorescent-activated cell sorting (FACS) and processed to determine sgRNAs frequencies after two rounds of selection, expansion, and enrichment. (**C**) Volcano plot depicting the Log2 (fold-change) and the negative Log10 (adjusted p-value) of the genes targeted by sgRNAs in 2DG-treated-sorted (TS) cells. This analysis identified genes whose loss confers repression of the ATF6α reporter during ER stress without impacting ER-stress-induced IRE1, compared to treated-unsorted (TUS) cells. The table lists the top 10 genes positively selected in the ATF6α *activator screen*, with FURIN highlighted as a hit for further analysis. (**D**) Volcano plot, as in 'C', depicting the genes targeted by sgRNAs in untreated-sorted (UTS) cells. This analysis identified genes whose loss confers activation of the ATF6α reporter under basal condition, compared to untreated-unsorted (UTUS) cells. The table lists the top 10 genes positively selected in the ATF6α *repressor screen*, with calreticulin (CRT) highlighted as the hit for further focussed analysis. (**H**) Venn diagram depicting unique and common upregulated top genes examined in 'D', between our screen (yellow) and previous high-throughput UPR screenings (*Adamson et al., 2016*; *Panganiban et al., 2019*; *Yang et al., 2018*) to discard hits that were confirmed to be IRE1- or PERK-dependent UPR target genes. The Log2 (fold-change) and the number of active sgRNA are provided for the 20 genes that remained in our screen, to be put forth as possible ATF6α repressor candidates.

The online version of this article includes the following source data and figure supplement(s) for figure 1:

**Figure supplement 1.** Characterisation of the ATF6α/IRE1 dual unfolded protein response (UPR) reporter cell (*XC45-6S*).

**Figure supplement 1—source data 1.** Raw images for gels shown in *Figure 1—figure supplement 1*.

**Figure supplement 2.** Enrichment method and quality control data analysis by MAGeCK of the CRISPR/Cas9 screens.

**Figure supplement 3.** Integration of CRISPR screen hits.

**Figure supplement 4.** FURIN depletion decreases responsiveness to ATF6α activation upon endoplasmic reticulum (ER) stress.

**Figure supplement 4—source data 1.** Raw images for gels shown in *Figure 1—figure supplement 4*.

*Figure 1—figure supplement 1B*). Additionally, the introduction of the reporter systems did not affect basal BiP protein levels when compared with plain CHO-K1 cells (*Figure 1—figure supplement 1C*).

Disruption of endogenous *cgATF6α* in *XC45-6S* cells by CRISPR/Cas9 genome editing abolished the responsiveness of the BiP::sfGFP reporter to ER stress and derepressed IRE1 signalling, both basally and in response to ER stress. These observations are consistent with ATF6α obligatory role in BiP expression and with findings that IRE1 signalling is repressed by ATF6α activity (*Walter et al., 2018*).

To further validate these UPR reporters, cells were subjected to tunicamycin treatment in the presence of Ceapin-A7 (a small molecule that blocks ATF6α activation by tethering it to the lysosome) (*Gallagher et al., 2016*) or an S1P inhibitor (PF429242) (*Hawkins et al., 2008*). Ceapin-A7 effectively blocked the activation of the ATF6α fluorescent reporter, whereas the S1P inhibitor partially attenuated the BiP::sfGFP signal in stressed cells (*Figure 1—figure supplement 1D*).

The selective IRE1 inhibitor 4µ8C (*Cross et al., 2012*) did not affect the ATF6α-dependent BiP::sfGFP reporter under ER stress conditions, but blocked the IRE1-dependent XBP1s::mCherry reporter (*Figure 1—figure supplement 1E*). Collectively, these observations confirm that the dual UPR reporter *XC45-6S* cell line enables independent monitoring of ATF6α and IRE1 activities, making it well suited for identifying additional modulators of ATF6α signalling by high-throughput techniques.

## Genome-wide CRISPR/Cas9 knock-out screens to profile ATF6α signalling

To identify regulators of ATF6α a library of pooled lentiviral single-guide RNAs (sgRNAs) targeting 20,680 predicted protein-coding genes in CHO cells, with ~6 sgRNAs per gene, was used (*Ordóñez et al., 2021*). Transduction of the *XC45-6S* ATF6α/IRE1 dual UPR reporter cells was performed at a coverage of ~640×, with a multiplicity of infection (MOI) of 0.3, intentionally set low to disfavour acquisition of more than one sgRNA by any one cell (*Shalem et al., 2014*). Non-transduced cells were purged by puromycin selection, and the pool of transduced cells was expanded for 10 days to favour gene editing and the establishment of loss-of-function phenotypes.

To systematically identify genes regulating the ATF6α transcriptional programme, two positive selection screens were conducted in parallel: a 'loss-of-ATF6α function screen' under ER stress conditions to identify activators/enhancers of ATF6α, and a 'gain-of-ATF6α function screen' under basal conditions to identify ATF6α repressors (*Figure 1B*). In the ATF6 *activator screen*, cells were stressed by exposure to 2-deoxy-D-glucose for 18 hr before fluorescent-activated cell sorting (FACS). Cells at the lowest 2% of BiP::sfGFP signal, with preserved IRE1 signalling (BiP::sfGFP$^{low}$; XBP1s::mCherry$^{high}$), were selected for further analysis. The search for genes whose inactivation derepresses ATF6α (ATF6α *repressor screen*) was conducted in the absence of ER stress. Here, cells exhibiting a constitutively active ATF6α reporter but only basal levels of the IRE1 reporter (BiP::sfGFP$^{high}$; XBP1s::mCherry$^{low}$) were selected for further analysis. Sorted cells from both screens underwent two rounds of progressive enrichment through cell culture expansion and phenotypic selection by sorting. As a result, a progressive increase in the proportion of BiP:sfGFP$^{low}$ or BiP:sfGFP$^{high}$ cells occurred in the respective screen (*Figure 1—figure supplement 2A* and *Figure 1—figure supplement 2B*).

The integrated sgRNAs were amplified from genomic DNA, and their abundance in the enriched populations after one or two rounds of sorting was estimated by deep sequencing and compared to a reference sample of transduced and unsorted population. MAGeCK bioinformatics analysis (*Li et al., 2014*) was used to determine sgRNA sequence enrichment, establishing the corresponding ranking of targeted genes (*Supplementary file 1*; see GEO accession number: GSE254745) and quality controls measures (*Figure 1—figure supplement 2C* and *Figure 1—figure supplement 2D*). Both screens were conducted at least in duplicate for all conditions.

## Expected and new candidate genes that contribute to ATF6α activation

MAGeCK-based analysis of sgRNAs enriched in cells with attenuated ATF6α signalling and intact activity of the IRE1 reporter, revealed that those targeting S1P and S2P proteases (encoding by the *MBTPS1* and *MBTPS2* genes) were the two most enriched genes (*Figure 1C*), validating the experimental methodology. Gene Ontology (GO) cluster analysis of the 100 most significantly enriched targeted genes revealed up to 19 enriched pathways (*Figure 1—figure supplement 3A*), with the top 2 pathways highlighted in *Figure 1C* being closely related to ATF6α signalling.

We focussed our attention on targeted genes that could act upstream to find early regulatory events of ATF6α. Therefore, targeted genes that were likely to act downstream and impact on the cleaved ATF6α form (N-ATF6α) were dismissed, including the *SP2* transcription factor, which shares the binding site with NF-Y (*Völkel et al., 2015*), and also genes involved in nuclear import, such as *NUP50* and *SEH1L* (*Figure 1C* and *Figure 1—figure supplement 3A*). Similar consideration was applied to *PRMT8*, as the family member *PRMT1* has been shown to enhance ATF6α transcriptional activity (*Baumeister et al., 2005*). Additionally, guides targeting *IGFR1* and *CXXC1* were also dismissed as likely affecting ER proteostasis indirectly via cellular growth (*Pfaffenbach et al., 2012*). Guides targeting the genes encoding the ER-resident proteins *SEC61A1*, *SEC61B* (both components of the translocon machinery), and the *DNAJB9*/ERDj4 co-chaperone were highly enriched. However, given their roles in selectively regulating IRE1 signalling (*Adamson et al., 2016*; *Shoulders et al., 2013*), we deemed their identification to reflect a bias towards IRE1 activators arising from the selection scheme for ATF6α$^{low}$ and IRE1$^{high}$ cells.

One of the genes that could plausibly play a proximal role and act upstream on ATF6α activation was the ubiquitous Golgi-localised protease FURIN (*Figure 1C*). Like S1P, FURIN belongs to the subtilisin-like proprotein convertase family (*Nakayama, 1997*; *Van de Ven et al., 1991*) and is thus poised to act at the level of ATF6α proteolytic processing. All six sgRNAs targeting *FURIN* were enriched in the BiP:sfGFP$^{low}$ cells, a level of enrichment similar to that observed to the established ATF6α regulators *MBTPS1* (S1P) and *MBTPS2* (S2P) (*Figure 1—figure supplement 3B*). The genotype–phenotype relationship suggested by the screen was also observed upon targeting *FURIN* in ATF6α/IRE1 dual reporter cells, as both pools of targeted cells and an individual clone had diminished responsiveness of BiP:sfGFP to ER stress (*Figure 1—figure supplement 4A*, left and middle panels). Furthermore, the S1P inhibitor more effectively attenuated the BiP::sfGFP reporter in *FURIN*Δ cells compared to wild-type (WT) cells (*Figure 1—figure supplement 4A*, right panel, *Figure 1—figure supplement 4B* and *Figure 1—figure supplement 4C*).

These observations raised the possibility of redundant roles of FURIN and S1P in activating ATF6α by cleaving its LD. However, ATF6α_LD expressed and purified from mammalian cells, failed to serve

as a substrate for FURIN *in vitro* (*Figure 1—figure supplement 4D*), suggesting instead that FURIN's role in regulating ATF6α signalling is indirect.

## Calreticulin and an interconnected gene network repress ATF6α

To identify genes that basally repress ATF6α, a similar MAGeCK-based data analysis was applied to the targeted genes enriched in cells that selectively activated the ATF6α reporter under basal conditions. GO analysis of the 100 most significantly enriched genes established an interconnected network highlighting signalling processes including the calreticulin (CRT)/calnexin (CNX) cycle, *N*-glycan biosynthesis, ERAD and COPI-dependent Golgi-to-ER retrograde traffic (*Figure 1D* and *Figure 1—figure supplement 3C*). Among these, the top 10 genes included CRT, components of the dolichol-phosphate mannose (DPM) synthase complex, such as *DPM3*, *DPM2*, and *DPM1* involved in the *N*-glycosylation process, and ERAD components, such as *UBE2G2* and *SEL1L* (*Figure 1D*).

Because these four pathways potentially activate all three branches of the UPR, genes previously identified in genome-wide screens to modulate IRE1 or PERK functionality were excluded from further analysis (e.g. *TMED10*, *SEL1L*, *FAB2*, *SYVN1*, and *PDIA6*) (*Figure 1H*; *Adamson et al., 2016*; *Panganiban et al., 2019*; *Yang et al., 2018*). Of the remaining genes, we focussed on those encoding proteins localised in the ER lumen or Golgi, which could act as specific proximal regulators of ATF6α. Among these, CRT, an abundant luminal ER lectin chaperone, caught our attention (*Figure 1H*). Notably, although CRT promotes the folding of ER synthesised glycoproteins *via* the CRT/CNX cycle (*Lamriben et al., 2016*), its ER transmembrane homologue CNX was not identified in the screen. This observation was further supported by the fact that sgRNAs targeting *CNX* were not enriched, whereas five out of six sgRNAs targeting *CRT* showed significant enrichment during the selection process (*Figure 2A*).

## Constitutive activation of ATF6α in cells lacking CRT

To explore *CRT*'s role in ATF6α regulation, *CRT* was targeted by CRISPR/Cas9 in WT *XC45-6S* dual UPR reporter cells. *CRT*-targeted cells exhibited large populations of BiP::sfGFP$^{high}$; XBP1s::mCherry$^{low}$ cells, mirroring the results of the initial screen. This phenotype was stable in single clones from the targeted pool (*Figure 2B, C*). Targeting CRT repressed the IRE1 reporter basally and left little room for further induction of the ATF6α reporter upon ER stress (*Figure 2B, C*). Derepression of the ATF6α reporter in *CRTΔ* clones correlated with a slight increase in BiP protein levels (*Figure 2D*). Immunoblots of ER-resident proteins (reactive with antibody directed to their endogenous KDEL C-terminal tag) confirmed the increase in BiP protein levels in *CRTΔ* cells and revealed no marked changes in GRP94 protein levels, a slight increase in P3H1 levels, and a notably increase in a band likely corresponding to PDIA in a *CRTΔ* clone (*Figure 2—figure supplement 1*). In contrast, depletion of CNX had no impact on the BiP::sfGFP reporter (*Figure 2—figure supplement 2A* and *Figure 2—figure supplement 2C*), consistent with the findings of the screen.

To test whether derepression of the BiP::sfGFP reporter in *CRTΔ* cells was ATF6α dependent, endogenous *ATF6α* was also inactivated in *CRTΔ* cells. ATF6α inactivation restored BiP::sfGFP reporter to its low baseline levels and derepressed the IRE1 reporter to levels observed in *XC45-6S* parental cells (*Figure 2E*). Overall, these findings confirmed that BiP::sfGFP reporter activity correlates with the transcriptional activity of ATF6α in *CRTΔ* cells.

## CRT represses Golgi-trafficking and processing of ATF6α

Considering the crucial role of trafficking in ATF6α activation, we next investigated whether depletion of CRT affected the stress-induced localisation of ATF6α by comparing the subcellular localisation of ATF6α in parental and *CRTΔ* cells. As trafficking is closely linked to proteolytic processing, WT ATF6α does not accumulate in the Golgi of stressed cells (*Shen et al., 2002*). To stabilise ATF6α molecules reaching the Golgi, a CHO-K1 cell line stably expressing a GFP-tagged version of *cg*ATF6α_LD lacking S1P and S2P cleavage sites (GFP-ATF6α_LD$^{S1P,S2Pmut}$) was engineered (*Figure 3A*). In non-stressed cells, GFPATF6α_LD$^{S1P,S2Pmut}$ was distributed in a prominent reticular ER pattern with only a minor fraction co-localising with the Scarlet-tagged Giantin Golgi localisation marker. Exposure to the rapidly acting ER stress-inducing agent dithiothreitol (DTT), increased the Golgi pool at the expense of the ER pool (*Figure 3A*); an expected outcome validating GFP-ATF6α_LD$^{S1P,S2Pmut}$ as a sentinel for the ATF6α protein. Compared to parental cells, GFP-ATF6α_LD$^{S1P,S2Pmut}$ showed significantly more Golgi co-localisation in *CRTΔ* cells under basal conditions (*Figure 3A*). These results were in line with our previous

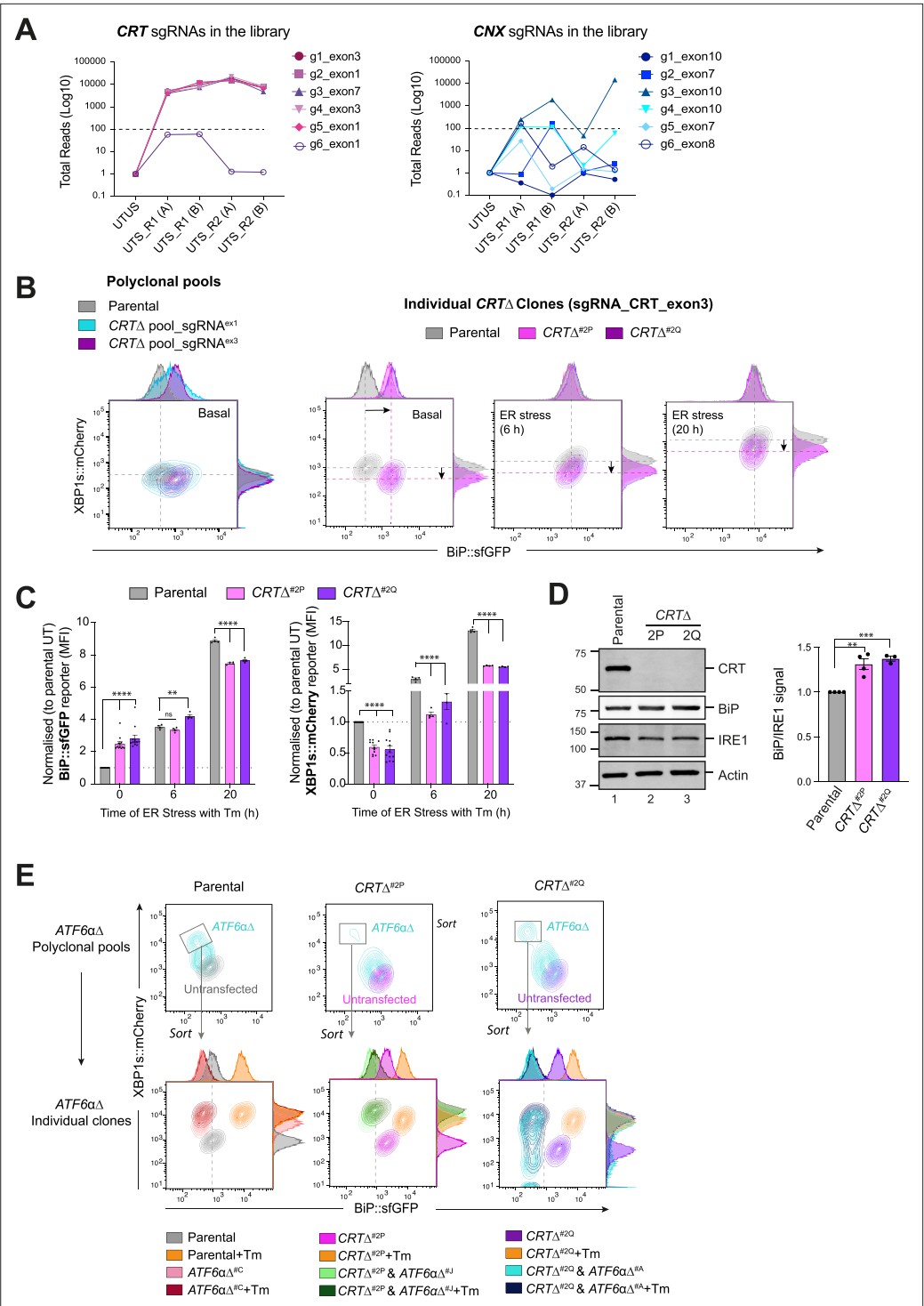

**Figure 2.** Calreticulin (CRT)-depleted cells exhibit constitutive activation of ATF6α signalling. (**A**) Total read count enrichment through the selection process for each active sgRNA targeting CRT and calnexin (CNX) [UTUS: untreated and unsorted; UTS: untreated and sorted; R1: round 1 of enrichment; R2: round 2 of enrichment; A–C: pools of cells; g1–g6: sgRNAs]. (**B**) Two-dimensional contour plots of BiP::sfGFP and XBP1s::mCherry signals examined in two derivative *CRTΔ* polyclonal pools (left) and two independent *CRTΔ* clones (right, named *CRTΔ*#2P and *CRTΔ*#2Q) under basal conditions or endoplasmic reticulum (ER) stress induced with 2.5 µg/ml Tm for a short (6 hr) and extended period of time (20 hr). A representative dataset from more than four independent experiments is shown. (**C**) Normalised quantification expressed as fold-change of the median fluorescence intensity (MFI) of BiP::sfGFP (left) and XBP1s::mCherry signals (right) in the two independent *CRTΔ* clones from more than

*Figure 2 continued on next page*

*Figure 2 continued*

four independent experiments, as described in 'B', indicated by mean ± standard error of the mean (SEM). (**D**) Representative immunoblots of endogenous CRT and BiP protein levels in cell lysates from *XC45-6S* parental cells and two *CRTΔ* derivatives clones selected for functional experiments. The samples were also blotted for IRE1 and actin (loading controls). The right graph bar shows the ratio of BiP to IRE1 signal in four independent experiments indicated by mean ± SEM. (**E**) *Top panel*: Two-dimensional contour plots of BiP::sfGFP and XBP1s::mCherry signals in polyclonal pools of parental cells and two *CRTΔ* clones after depleting the endogenous *ATF6α* locus by CRISPR/Cas9 gene editing. *Lower panel*: Grey rectangles in top panels indicate the subcellular polyclonal population re-sorted to analyse single *ATF6αΔ* clones in parental and *CRTΔ* clones. ER stress treatments with 2.5 µg/ml Tm lasted 20 hr. All statistical analysis was performed by two-sided unpaired Welch's *t*-test and significance is indicated by asterisks (**p < 0.01; ***p < 0.001; ****p < 0.0001).

The online version of this article includes the following source data and figure supplement(s) for figure 2:

**Source data 1.** Raw images for gels shown in *Figure 2*.

**Figure supplement 1.** ER-resident proteins (with a KDEL-COOH tag) in calreticulin (CRT)-depleted cells.

**Figure supplement 1—source data 1.** Raw images for gels shown in *Figure 2—figure supplement 1*.

**Figure supplement 2.** Calnexin and ERp18 depletion do not deregulate ATF6α pathway.

**Figure supplement 2—source data 1.** Raw images for gels shown in *Figure 2—figure supplement 2*.

flow cytometry observations and suggested that CRT contributes to ER retention of ATF6α, either directly or indirectly.

GFP-ATF6α_LD$^{S1P,S2Pmut}$ reporter cannot be proteolytically processed. Consequently, to gauge the effect of CRT on ATF6α processing we sought to create a CHO-K1 stable cell line to track endogenous ATF6α. For that, the *cgATF6α* locus was tagged with a 3xFLAG and a monomeric GreenLantern (mGL) fluorescent protein, generating a cell line referred to as *2K*: 3xFLAG-mGL-ATF6α (*Figure 3B*). The functionality of this probe was confirmed by treatment with DTT that resulted in the appearance of a prominent faster migration band in sodium dodecyl sulfate–polyacrylamide gel electrophoresis (SDS–PAGE), compatible with the processed S2P-cleaved ATF6α (N-ATF6α) form, and depletion of the precursor form (ATF6α-P) (*Figure 3B.1*, lane 8). Subsequently, *2K* cells were targeted to generate *2K-CRTΔ* derivatives cells, enabling a comparison of ATF6α processing. *2K-CRTΔ* clonal cells exhibited elevated levels of the processed N-ATF6α form under both basal and stress-induced conditions (*Figure 3B.1,B.2*) that correlated with a higher nuclear mGL-ATF6α signal observed by live cell microscopy (*Figure 3C*). These findings indicated that CRT represses both ER-to-Golgi trafficking and processing of ATF6α into its active form.

It has been reported that ATF6α can exist in three redox forms: a monomer and two inter-chain disulfide-stabilised dimers (*Nadanaka et al., 2007*). To investigate whether CRT depletion alters the redox status of ATF6α, cell lysates from *2K*-parental cells and *2K-CRTΔ* clonal cells were analysed under both basal and ER-stress conditions using non-reducing SDS–PAGE (*Figure 3—figure supplement 1A*). We detected two redox forms of ATF6α, consistent with an inter-chain disulfide-stabilised dimer and a monomer. Under basal conditions, ATF6α predominantly existed as a high mobility monomer in both *2K*-parental and *2K-CRTΔ* cells, with the appearance of an even faster-migrating processed N-ATF6α form in *2K-CRTΔ* cells, consistent with the findings presented in *Figure 3B*. ER stress (induced by 2DG) was associated with appearance of a low-mobility band consistent with a disulfide-stabilised dimer, and a concomitant decrease in the ATF6α monomer intensity in *2K*-parental cells, as previously reported (*Oka et al., 2022*). Interestingly, under stress conditions, the monomer levels in *2K-CRTΔ* cells remained largely unchanged, indicating a higher monomer-to-dimer ratio compared to parental cells (*Figure 3—figure supplement 1B*). These data suggest that the loss of CRT may stabilise the monomeric form of ATF6α, which is proposed to be more efficiently trafficked. This observation aligns with our results showing that CRT depletion is linked to activation of ATF6α.

## CRT interacts with the ATF6α LD *in vitro*

To investigate the underlying mechanism of CRT's repressive effect on ATF6α, we explored the physical interaction between the two purified proteins *in vitro* using BioLayer Interferometry (BLI). The biotinylated ATF6α_LD, expressed in mammalian cells, was immobilised on the BLI probe and used as the ligand, and bacterially expressed CRT was used as the analyte. Considering possible regulatory

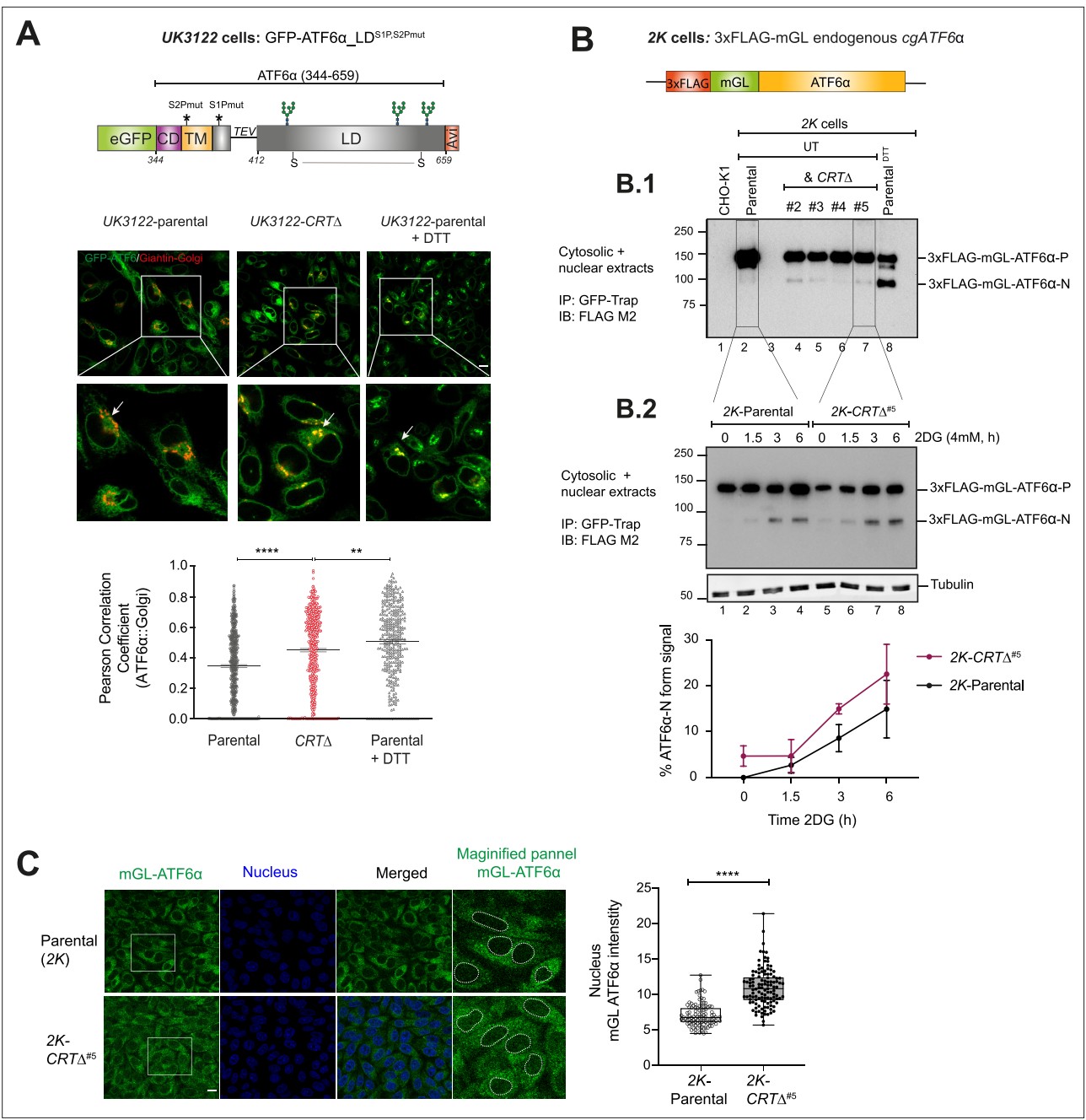

**Figure 3.** Loss of calreticulin promotes higher proportion of ATF6α trafficking to Golgi and nucleus. (**A**) *Top panel*: Schematic representation of the genetic construct (UK3122) designed for generating a CHO-K1 stable cell line expressing a N-terminus-tagged ATF6α, where eGFP replaces most of the cytosolic domain of *cg*ATF6α, and the S2P and S1P cleavage sites are mutated to increase the intact fraction in the Golgi. Colour code: eGFP tag in green; cytosolic domain (CD) in purple; transmembrane domain (TM) in yellow; Tobacco Etch Virus (TEV) protease cleavage site (ENLYFQ↓G) is represented with a linker line; luminal domain (LD) in grey; and a C-terminus Avi-tag in red. *Lower panels*: Representative live cell confocal microscopy images of *UK3122* CHO-K1 cells stably expressing GFP-ATF6α_LD^S1P,S2Pmut and transiently expressing a pmScarlet _Giantin-C1 plasmid as a red fluorescent Golgi marker. Both parental cells and a CRISPR/Cas9 gene edited *CRTΔ* derivative pool were imaged. Dithiothreitol (DTT) treatment (4 mM, 1 hr) was applied to parental cells to positively influence ATF6α's Golgi localisation. Each panel's magnified sections are presented in the insets. Arrows indicate GFP-ATF6α_LD^S1P,S2Pmut::Golgi co-localisation. Scale bar: 10μm. Pearson coefficients for the co-localisation of GFP-ATF6α_LD^S1P,S2Pmut with the Golgi apparatus marker Giantin in parental cells (*n* > 150), parental cells treated with 4 mM DTT for 1 hr (*n* > 100) and *CRTΔ* cells (*n* > 150) are presented in the bar graph, including cells from three independent experiments. Volocity software was used for co-localisation quantification. Statistical analysis was performed by a two-sided unpaired Welch's *t*-test and significance is indicated by asterisks (**p < 0.01; ****p < 0.0001). (**B**) Schematic representation of engineered stable *2K* cells with the endogenous *cg*ATF6α locus (in orange) tagged with 3xFLAG (in red) and mGreenLantern (mGL) (in green) at the N-terminus. (**B.1**) Cell lysates from *2K* cells and four independent *2K-CRTΔ* derivative clones were harvested, immunoprecipitated (IP) using GFP-Trap

*Figure 3 continued on next page*

*Figure 3 continued*

Agarose (ChromoTek) and analysed by immunoblot (IB) using an anti-FLAG M2 antibody to detect processing of ATF6α. Full-length ATF6α-Precursor (ATF6α-P) and processed ATF6α-N forms were identified. Treatment with DTT for 1 hr (4 mM) was used to induce ATF6α processing. Parental untagged cells were used as control for background. Data shown are representative of one experiment. (**B.2**) Parental *2K* cells and a derivative single *CRTΔ* clone (#5) were treated with 4 mM 2-deoxy-D-glucose (2DG) at the indicated time points (1.5, 3, and 6 hr). Cells were harvested and analysed by immunoblot as in 'B.1'. Lower panel shows the percentage of ATF6α-N form signal/total ATF6α signal from three independent experiments indicated by mean ± standard error of the mean (SEM). (**C**) ATF6α nuclear translocation live cell microscopy assay showing unstressed *2K* parental cells and a *2K-CRTΔ* clone. mGL-ATF6α is in green, and nucleus in blue. mGL-ATF6α signal intensity in the nucleus (shown by dashed lines) was measured using Volocity software. Scale bar: 10μm. Data from parental cells (*n* > 100) and *CRTΔ* cells (*n* > 100) are displayed in a box and whiskers graph, displaying all points (with min. and max. intensities).

The online version of this article includes the following source data, source code, and figure supplement(s) for figure 3:

**Source data 1.** Raw images for gels shown in *Figure 3*.

**Figure supplement 1.** Analyses of ATF6α redox status in calreticulin (CRT)-depleted cells.

**Figure supplement 1—source code 1.** Raw images for gels shown in *Figure 3—figure supplement 1*.

roles of *N*-glycosylation and disulphide bonds in ATF6α activation (*Hong et al., 2004*; *Oka et al., 2019*), three different versions of ATF6α_LD were used: (1) fully glycosylated ATF6α_LD (WT), (2) non-glycosylated ATF6α_LD (ΔGly), and (3) ATF6α_LD lacking its cysteines (ΔC) (*Figure 4A*). BLI showed reversible binding of CRT to ATF6α_LD$^{WT}$ that was characterised by slow association ($k_{on}$) and dissociation ($k_{off}$) rates (*Figure 4B*). Both the association and dissociation phases were fitted to a single exponential association-dissociation model, yielding a $K_D$ within the range of 1 μM, a concentration that was consisted with the estimated CRT concentration in the ER of CHO-K1 cells obtained through quantitative immunoblotting (~7 μM) (*Figure 4—figure supplement 1*). Similar interactions were also observed with ATF6α_LD$^{ΔGly}$ and ATF6α_LD$^{ΔC}$.

To examine a possible interaction between CRT and ATF6α in cells, HEK293T cells were transfected with either GFP-ATF6α_LD$^{WT}$ or GFP-ATF6α_LD$^{ΔGly}$, followed by selective recovery using GFP-Trap Agarose and subsequent immunoblotting with an anti-CRT antibody. The co-immunoprecipitation results showed that both ATF6α_LD$^{WT}$ and, to a lesser extent, ATF6α_LD$^{ΔGly}$ immunoprecipitated endogenous CRT (*Figure 4C*). Re-probing the blot with an anti-GFP antibody revealed that GFP-ATF6_LD$^{ΔGly}$ was expressed at lower levels than ATF6α_LD$^{WT}$ (*Figure 4C*), potentially accounting for the reduced recovery of CRT in complex with ATF6α_LD$^{ΔGly}$. Overall, these observations suggested that CRT binds to ATF6α_LD both in cells and in isolation *in vitro*, supporting the notion of a direct interaction between the two proteins.

## A genetic platform to study endogenous ATF6α signalling

To circumvent the potentially corrupting effect of protein overexpression, we sought to express ATF6α variants from the endogenous locus, by replacing the ATF6αLD-coding sequence with mutants of our design. To enable CRISPR–Cas9-mediated homology-directed repair (HDR) that replaces the WT sequence with the mutants, we first created a non-functional ATF6α allele that could be subsequently restored to function by homologous recombination, offering WT or mutant repair templates (*Figure 5A*). As expected, the deletion of ATF6α resulted in the loss of the BiP::sfGFP signal in stressed cells (and enhanced XBP1s::mCherry signal in basal conditions, *Figure 5A*). This ATF6αΔ clone was re-targeted with a unique sgRNA/Cas9, alongside ATF6α_LD$^{WT}$, ATF6α_LD$^{ΔGly}$, or ATF6α_LD$^{ΔC}$ repair templates (*Figure 5B*).

Unlike the ATF6α_LD$^{WT}$, neither the ATF6α_LD$^{ΔGly}$ nor the ATF6α_LD$^{ΔC}$ repair templates fully restored BiP::sfGFP responsiveness to stress (*Figure 5B*, top panel), consistent with the importance of *N*-glycosylation and disulphide bond formation to ATF6α functionality. Notably, whereas ATF6α_LD$^{ΔC}$ knock-in cells exhibited basal activation of the IRE1 reporter comparable to levels observed in the ATF6αΔ parent (*Figure 5B*, lower panel), the IRE1 reporter signal in the ATF6α_LD$^{ΔGly}$ knock-in cells shifted to that observed in WT cells, suggesting that ΔGly allele retained some functionality. The partial ATF6α functionality of the ATF6α_LD$^{ΔGly}$ knock-in allele aligned with the lower expression levels of ATF6α_LD$^{ΔGly}$ compared to ATF6αLD$^{WT}$ in transfected cells, as previously observed in *Figure 4C*. However, this feature complicated interpretation of experiments designed to establishing the role of ATF6α_LD glycans on CRT-mediated repression of ATF6α: failure of CRT deletion to derepress the BiP::sfGFP signal in ATF6α_LD$^{ΔGly}$ knock-in cells (*Figure 5—figure supplement 1*) could be

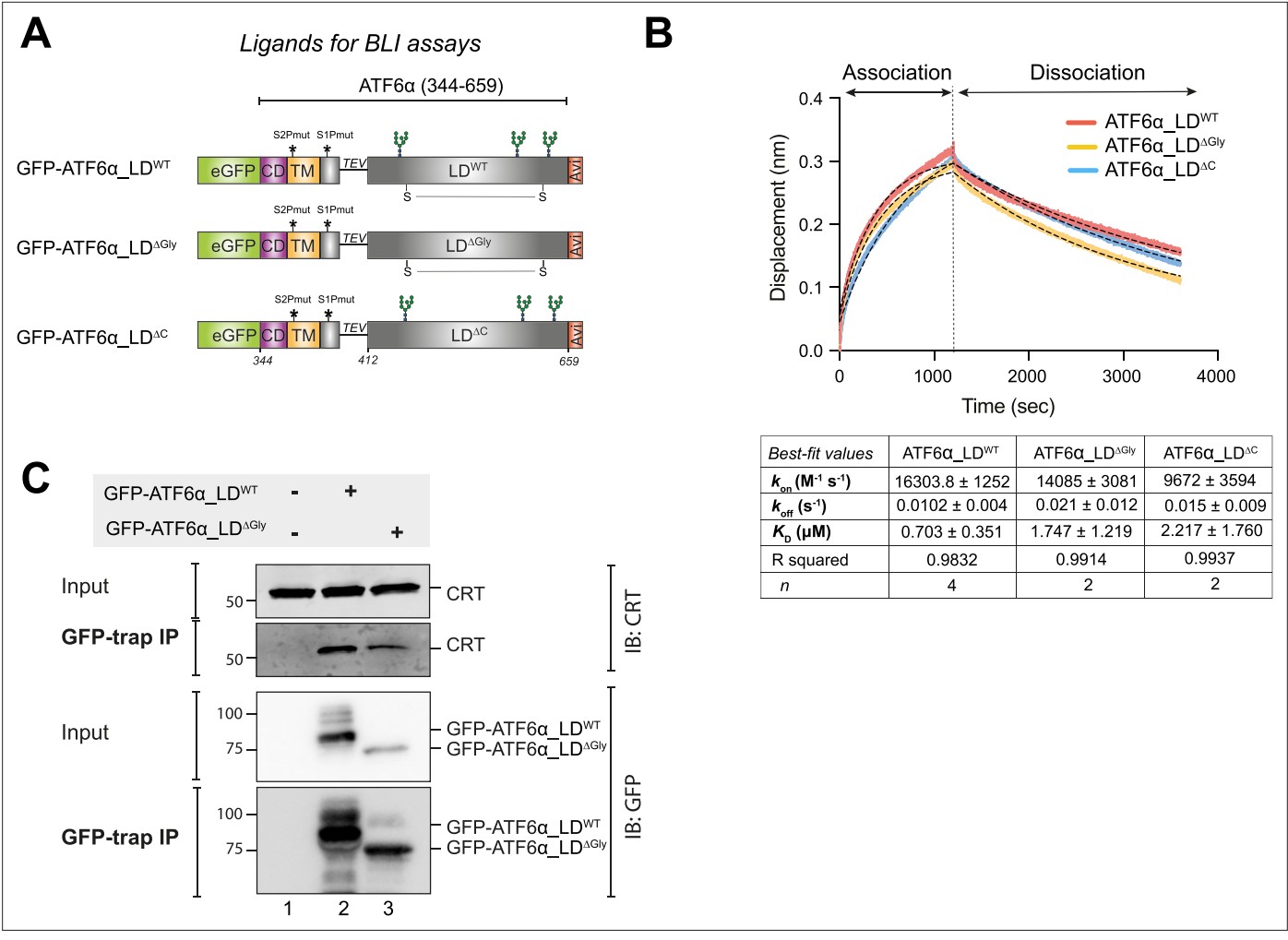

**Figure 4.** Calreticulin (CRT) interacts directly with the ATF6α luminal domain (LD) *in vitro* and in cells. (**A**) Schematic representation of the constructs designed to produce three LDs of ATF6α to be used as ligands in biolayer interferometry (BLI) assays: wild-type (WT) ATF6α_LD, non-glycosylated (ΔGly) ATF6α_LD, and ATF6α_LD lacking its cysteines (ΔC). The probes underwent *in vivo* biotinylation, purification using GFP-Trap Agarose (ChromoTek), and were eluted via TEV cleavage. Colour code: eGFP tag in green; cytosolic domain (CD) in purple; transmembrane domain (TM) in yellow; TEV cleavage site with a linker line; LD in grey; C-terminus Avi-tag in red. S–S indicates for the disulphide bond in the LD between two cysteines. (**B**) Representative BLI signals depicting the interaction of rat (r) CRT (analyte) with different forms of *cg*ATF6α_LD (from HEK293T cells) immobilised on streptavidin biosensors: ATF6α_LD^WT in red, ATF6α_LD^ΔGly in yellow, and ATF6α_LD^ΔC in blue. The interaction kinetics were calculated over the entire 1200s of the association phase and 2400s of the dissociation phase. Dotted lines represent the best-fit association then dissociation curves using a nonlinear regression to calculate kinetic constants by Prism10. The table represents the BLI measurements of the kinetics of the interaction between rCRT and *cg*ATF6α_LD from two to four independent experiments (means ± standard error of the mean [SEM]). Interaction kinetics were measured at an rCRT concentration of 10 μM. (**C**) Representative immunoblots (IB) of endogenous CRT recovered in complex with ATF6α in HEK293T cells overexpressing GFP-*cg*ATF6α_LD^WT, GFP-*cg*ATF6α_LD^ΔGly, or neither. Immunoprecipitations (IP) were performed using GFP-Trap Agarose, and then membranes were blotted with anti-human CRT and anti-GFP antibodies. The immunoblots are representative for three independent experiments.

The online version of this article includes the following source data and figure supplement(s) for figure 4:

**Source data 1.** Raw images for gels shown in *Figure 4*.

**Figure supplement 1.** Estimation of calreticulin's (CRT) physiological abundance by quantitative immunoblotting.

**Figure supplement 1—source data 1.** Raw images for gels shown in *Figure 4—figure supplement 1*.

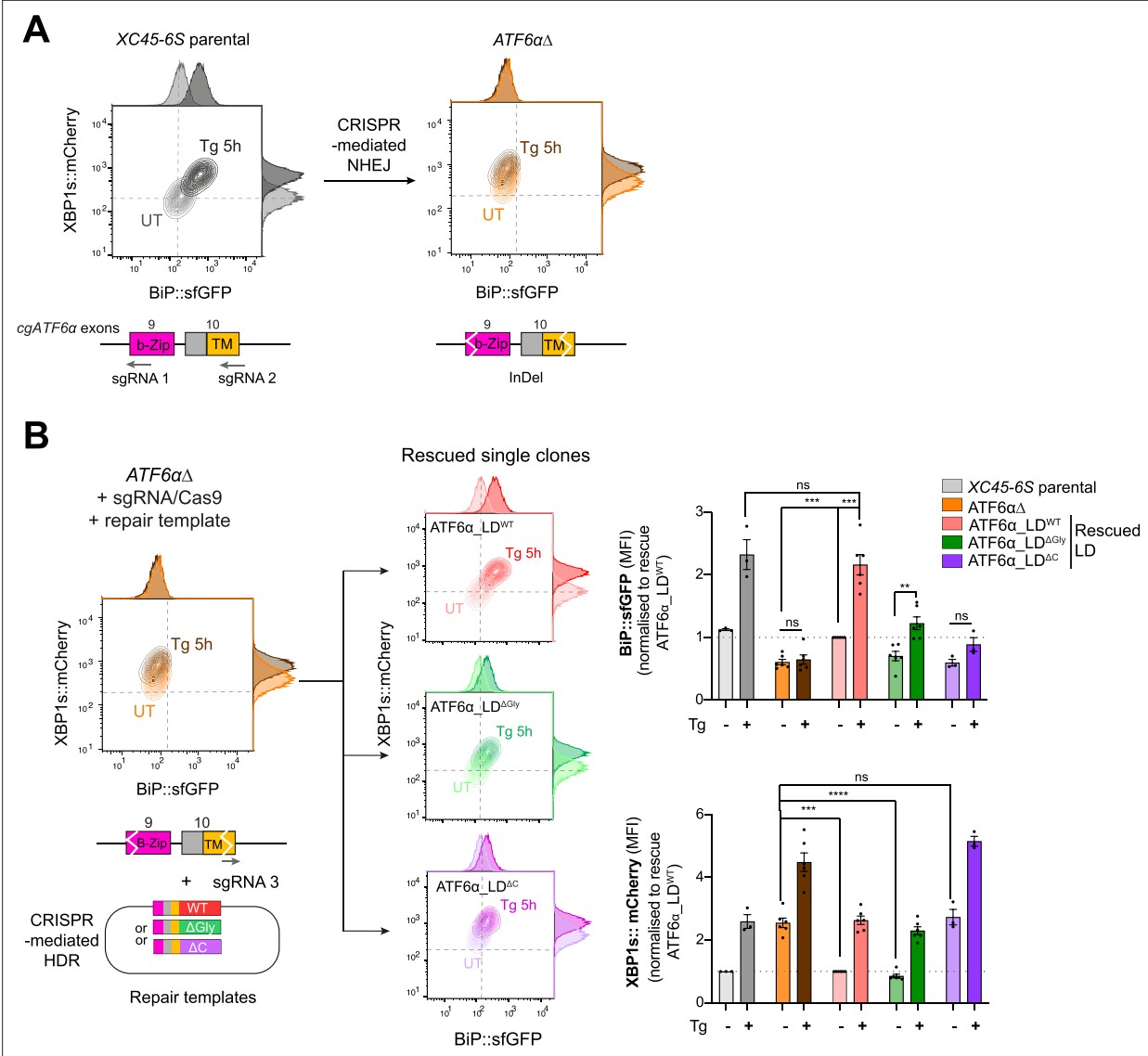

**Figure 5.** A homologous CRISPR/Cas9 gene editing recombination platform to study ATF6α variants at endogenous levels in CHO-K1 cells. (**A**) Two-dimensional contour plots of BiP::sfGFP and XBP1s::mCherry signals in *XC45-6S* parental cells (in grey; left) and in a null functional ATF6α clonal cell (*ATF6αΔ*, in orange; right) generated by CRISPR/Cas9-mediated NHEJ (non-homologous end joining) to introduce frameshifting mutations into the DNA-binding (b-Zip, pink) and transmembrane (TM, yellow) encoding regions by targeting exons 9 and 10 of the endogenous locus of *cgATF6α*. Cells were analysed in basal conditions (UT) or upon endoplasmic reticulum (ER) induction with Tg (0.5 μM, 5 hr). A schema for the sgRNAs/Cas9 that target cgATF6 is shown below the plots. (**B**) *ATF6αΔ* cells were re-targeted with a sgRNA/Cas9 directed to the mutated exon 10 and offering three repair templates: ATF6α_LD^WT in red, ATF6α_LD^ΔGly in green, and ATF6α_LD^ΔC in purple. Successfully rescued single clones by CRISPR/Cas9-mediated homology-directed repair (HDR) were evaluated in the absence (pale colour) and presence of ER stress (0.5 μM Tg, 5 hr, dark colour). Contour plots are representative from three to six independent experiments. Bar graphs display the quantification of the fold-change of BiP::sfGFP (top) and XBP1s::mCherry signals (bottom) in more than three independent experiments indicated by mean ± standard error of the mean (SEM). Statistical analysis was performed by a two-sided unpaired Welch's *t*-test and significance is indicated by asterisks (***p < 0.001; ****p < 0.0001; ns: non-significant).

The online version of this article includes the following figure supplement(s) for figure 5:

**Figure supplement 1.** Calreticulin (CRT) depletion in ATF6α_LD^ΔGly knock-in cells does not derepress basally ATF6α reporter activity.

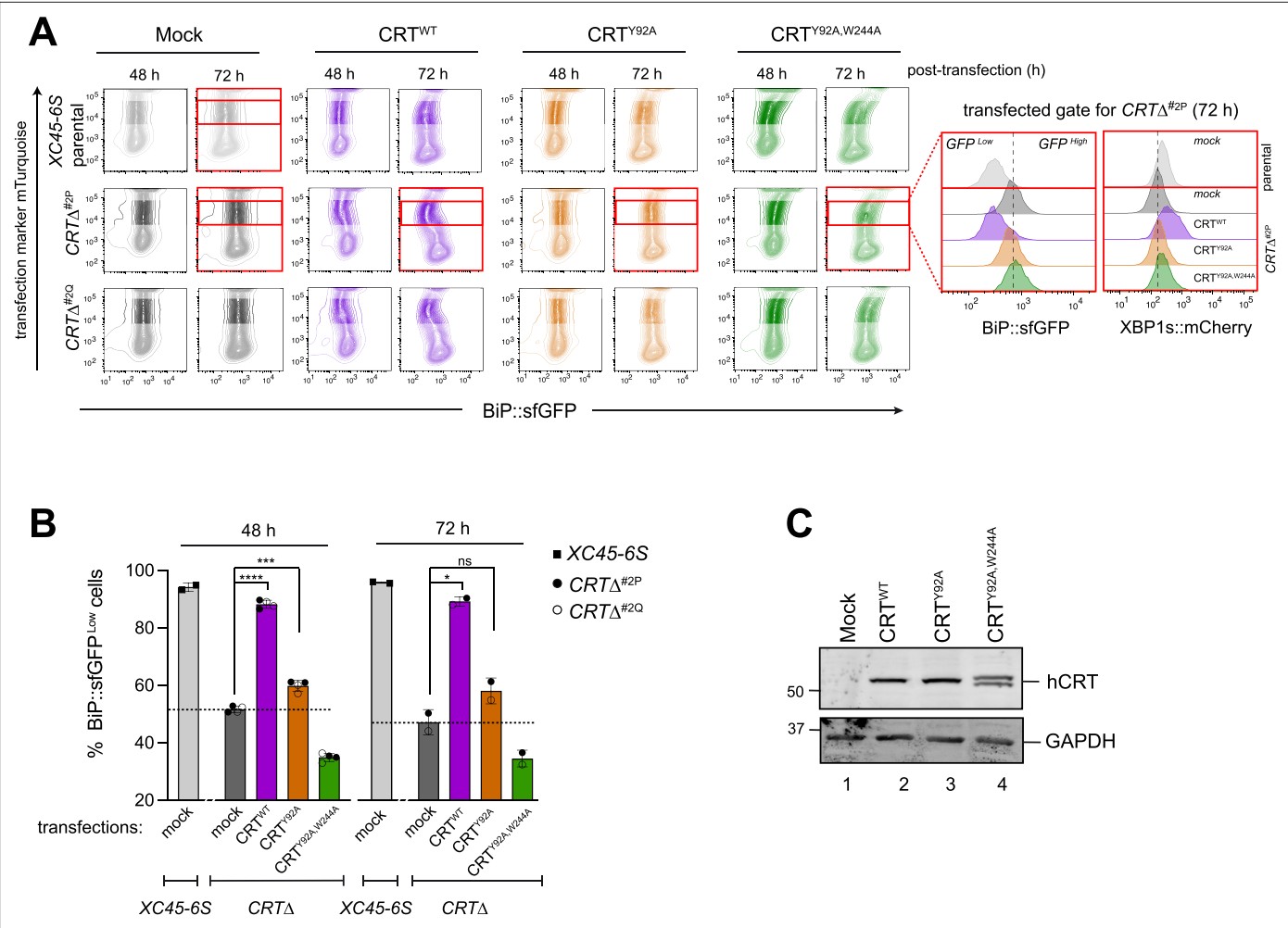

**Figure 6.** Impact of calreticulin (CRT) overexpression on cells depleted of CRT. (**A**) Two-dimensional contour plots of BiP::sfGFP and mTurquoise signals (transfection marker) in parental cells and two derivative *CRTΔ* clones transiently transfected for 48 and 72 hr with plasmids encoding wild-type human CRT (CRT[WT]) or two lectin CRT mutants, CRT[Y92A] and CRT[Y92A,W244A]. As a control for transfection, cells were transfected with a mock plasmid. Shadowed rectangles mark cells that were not selected for analysis because they had low levels of mTurquoise-tagged plasmid (indicating low transfection) or have very high transfection levels that could be toxic. Cells within unshadowed rectangles and marked with red delineate those cells expressing moderate/high levels of mTurquoise-tagged plasmid selected for analysis of BiP::sfGFP and XBP1s::mCherry signals distribution. Right histograms indicate the transfected, gated cells 72 hr post-transfection (red rectangles). (**B**) Bar graph displaying the percentage of BiP::sfGFP[low] cells (rescue phenotype) in mTurquoise-positive cells gated by the unshadowed boxes in 'A' as mean ± standard deviation (SD) of data obtained from one (72 hr) to two (48 hr) independent experiments. Data from both *CRTΔ* clones have been combined in each time point of analysis. Statistical analysis was performed by a two-sided unpaired Welch's *t*-test and significance is indicated by asterisks (*p < 0.05; ***p < 0.001; ****p < 0.0001; ns: non-significant). (**C**) CHO-K1 cells were transiently transfected with the hCRT plasmids used in 'A' for 72 hr. Following transfection, cell lysates were recovered, and equal protein amounts were loaded into 12.5% sodium dodecyl sulfate–polyacrylamide gel electrophoresis (SDS–PAGE) gels and immunoblotted for hCRT to analyse protein stability in CRT mutants. GAPDH served as a loading control. The immunoblots are representative for two independent experiments.

The online version of this article includes the following source data for figure 6:

**Source data 1.** Raw images for gels shown in *Figure 6*.

interpreted as either an obligatory role for glycans in the repression or as lack of dynamic range of the signal needed to detect a residual effect of CRT deletion in ATF6α_LD[ΔGly] knock-in cells.

## Impact of CRT variants on ATF6α reporter activity

CRT is a lectin chaperone that interacts with the PDI ERp57 to stabilise CRT–substrate binding (*Oliver et al., 1999*). Therefore, we tested the ability of human CRT[WT] and two mutants that compromised glycan binding (Y92A) or Erp57 co-chaperone binding (W244A) (*Del Cid et al., 2010*) to reverse the *CRTΔ* phenotype. Parental and *CRTΔ* clones were transfected with CRT[WT], CRT[Y92A], and the double

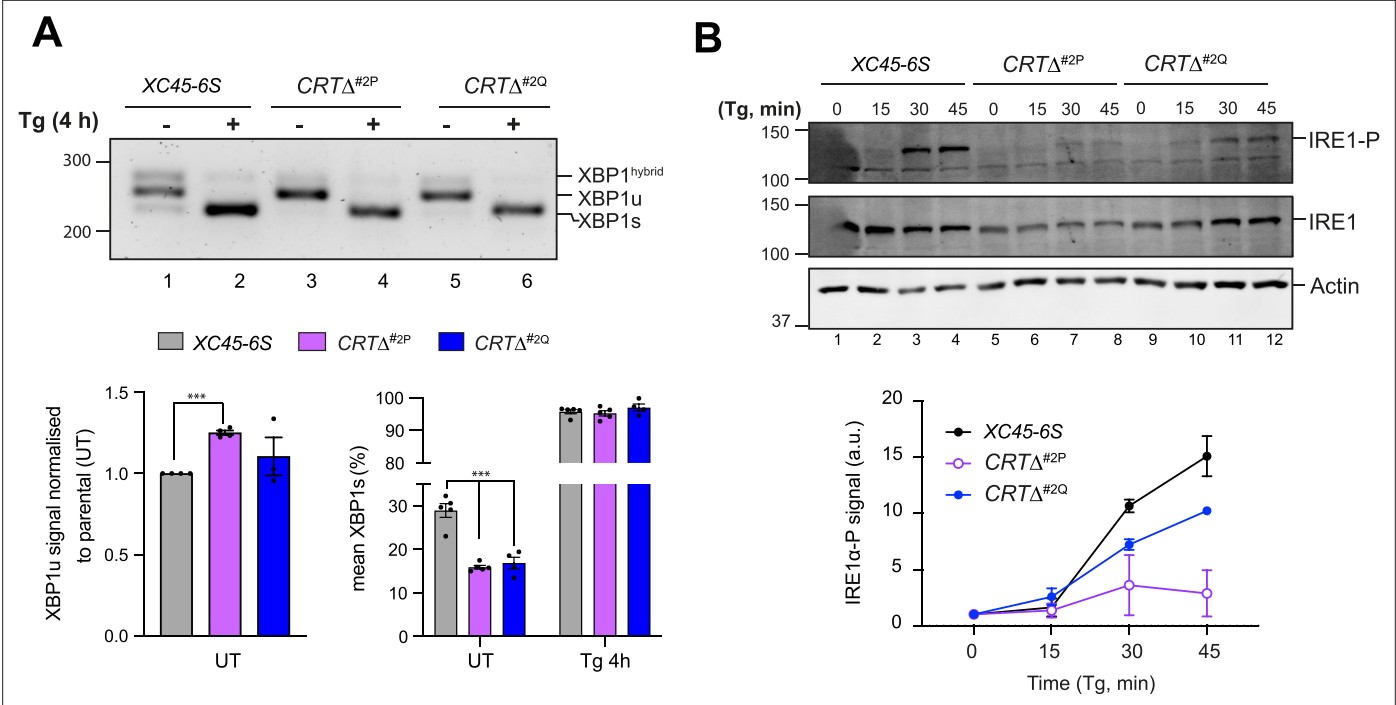

**Figure 7.** Calreticulin depletion exposes a negative feedback loop of ATF6α activity on IRE1. (**A**) *Top panel*: Agarose gel of XBP1 cDNA from thapsigargin (Tg)-treated parental cells and two *CRTΔ* clones. Migration of the unspliced (XBP1u), spliced XBP1 (XBP1s), and hybrid (XBP1^hybrid) stained DNA fragments is indicated. The XBP1^hybrid band represents DNA species containing one strand of XBP1u and one strand of XBPs (***Shang and Lehrman, 2004***). Splicing was assessed by reverse transcription (RT)-PCR. *Lower panel*: Plots displaying the total XBP1u signal in the left panel, and the fraction of XBP1s in the right panel. For quantification, 50% of the hybrid band signal was assumed to be XBP1s and the other 50% XBP1u. The bars and error bars represent the mean ± standard error of the mean (SEM) of data obtained from more than three independent experiments. Statistical analysis was performed by a two-sided unpaired Welch's *t*-test and significance is indicated by asterisks (***p < 0.001). (**B**) Representative immunoblots of endogenous levels of cgIRE1 and phosphorylated IRE1 (IRE1-P) in parental cells and two *CRTΔ* clones basally and upon endoplasmic reticulum (ER) stress induction with 0.5 µM Tg for the indicated times. The levels of IRE1-P signal over the time in each genotype are represented in a graph as mean ± standard deviation (SD) of data obtained from two independent experiments.

The online version of this article includes the following source data for figure 7:

**Source data 1.** Raw images for gels shown in *Figure 7*.

mutant CRT^Y92A,W244A expressing plasmids, and cells were evaluated by flow cytometry 48 and 72 hr post-transfection (***Figure 6A***). Expression of CRT^WT restored basal levels of both BiP::sfGFP and XBP1s:mCherry reporters in *CRTΔ* cells (***Figure 6A***, purple panels and ***Figure 6B***), confirming the repressive role of CRT on ATF6α. In contrast, expression of the double CRT^Y92A,W244A mutant failed to rescue the *CRTΔ* phenotype (***Figure 6A***, green panels and ***Figure 6B***), whereas CRT^Y92A, selectively incapable of glycan binding, retained some ability to repress BiP::sfGFP and modestly rescuing the *CRTΔ* phenotype (***Figure 6A***, orange panels and ***Figure 6B***). Immunoblots targeting CRT raised concerns about the integrity of theCRT^Y92A,W244A protein, precluding a separate analysis of Erp57's role in ATF6α repression (***Figure 6C***). However, CRT^WT and CRT^Y92A proteins were expressed similarly (***Figure 6C***). Together, our observations pointed to an important role for interactions between CRT and ATF6α_LD glycans but also left room for some lectin independent repression of ATF6α by CRT.

## CRT depletion exposes a negative feedback loop by which ATF6α represses IRE1

Given the well-established interplay between the ATF6α and IRE1 branches of the UPR, we sought to assess whether the constitutive activation of ATF6α observed in *CRTΔ* clones could be a contributing factor to the conspicuous downregulation of IRE1 activity. Through reverse transcription (RT)-Polymerase Chain Reaction (PCR) and gel electrophoresis, IRE1-mediated XBP1 mRNA splicing in parental cells and *CRTΔ* clones were compared. Under basal conditions, unspliced mRNA levels

(XBP1u) were significantly higher in *CRTΔ* clones (*Figure 7A*, left bar graph), consistent with the role of NATF6α in the transcriptional activation of XBP1 (*Yoshida et al., 2001*). In addition, compared to parental cells, *CRTΔ* cells exhibited a smaller fraction of spliced XBP1 mRNA (XBP1s) (*Figure 7A*, right bar graph), consistent with the lower IRE1 activity measured by a XBP1s::mCherry reporter.

IRE1 activity is closely tied to its phosphorylation state (*Shamu and Walter, 1996*), therefore IRE1α-phosphorylation (IRE1α-P) levels were compared in parental and *CRTΔ* clones. Accordingly, *CRTΔ* cells exhibited lower levels of stress-induced phosphorylation of their endogenous IRE1 protein and slightly reduced total IRE1 levels (*Figure 7B*). These results supported earlier findings that IRE1 signalling is repressed by ATF6α activity (*Walter et al., 2018*). However, the defect in IRE1 activity in CRT-depleted cells was partial, as in the presence of ER stress, all genotypes exhibited a comparable increase in XBP1s (*Figure 7A*).

## Discussion

This study reports on a comprehensive and unbiased genome-wide CRISPR/Cas9-based knockout screen dedicated to discovering regulators of ATF6α. Whereas previous high-throughput studies have broadly studied modulators of the UPR (*Adamson et al., 2016*; *Jonikas et al., 2009*; *Panganiban et al., 2019*), we designed an ATF6α/IRE1 dual UPR reporter cell line to selectively detect regulators of ATF6α. Although a connection between CRT and ATF6α was proposed previously based on an interaction between overexpressed proteins (*Hong et al., 2004*), our study utilises unbiased genetic tools and manipulation of endogenous proteins to establish CRT's role as a selective repressor of endogenous ATF6α signalling, mapping its function in retaining ATF6α in the ER.

It is notable that, though the screen identified well-known regulators of ATF6α processing (including S1P, S2P, and NF-Y, and thus validating the experimental approach), only a limited number of new genes that contribute to ATF6α signalling emerged from the screen. Despite the biochemical evidence for the role of COPII-coated vesicles and the oligomeric state of ATF6α in its activation, no cluster of genes involved in COPII vesicular transport or PDIs were found among the most enriched pathways. Nor were any counterparts to SCAP or INSIG, regulators of ER-to-Golgi trafficking of the related SREBP proteins (*Yang et al., 2002*), identified. This might reflect essentiality of the genes involved or redundancy amongst them. Cell-type specificity may also be implicated in failure to identify components that are unimportant to ATF6α activation in CHO-K1 cells. For example, deletion of ERp18, a small PDI-like protein implicated in ATF6α activation in HEK cells (*Oka et al., 2019*), affected neither the ATF6α nor IRE1α UPR branches in CHO-K1 cells studied here (*Figure 2—figure supplement 2D*).

The screen implicated two components of the nuclear pore complex (*NUP50* and *SEH1L*) in ATF6α pathway activity. While it might be tempting to consider a role for these components in the nuclear translocation of the processed soluble ATF6α N-terminal domain, there is no reason to think it would be regulatory, as it operates downstream of ER stress-regulated trafficking and processing events.

In addition to the identified S1P and S2P proteases, the *screen for ATF6α activators* identified FURIN, a Golgi-localised protease belonging to the subtilisin-like proprotein convertase family (like S1P protease) responsible for proteolytic activation of a wide array of precursor proteins within the secretory pathway (*Braun and Sauter, 2019*). FURIN deletion synergised with S1P inhibitors in attenuating ATF6α signalling. Additionally, ATF6α_LD has a conserved region (RTK̲S̲R̲R) related to the FURIN cleavage motif generally described as RX-R-X-[K/R]-R↓ (*Nakayama, 1997*; *Figure 1—figure supplement 4E*). It has been suggested that S1P's role in activation is to reduce the size of ATF6α's_ LD, thereby promoting S2P cleavage (*Shen et al., 2002*; *Ye et al., 2000*). FURIN cleavage could contribute to such a process and account for partial redundancy between S1P inhibition and FURIN depletion observed here. Nonetheless, *in vitro* FURIN failed to cleave the ATF6α_LD. These findings collectively suggest that FURIN may function as an indirect enhancer of ATF6α signalling, perhaps by setting up conditions in a post-ER compartment that favours ATF6α cleavage and activation.

The *screen for ATF6α repressors* revealed a substantial enrichment of genes involved in glycoprotein metabolism. Among that cluster, the most notable hit, not previously identified in other high-throughput UPR screens, was CRT, a well-characterised soluble ER chaperone with a key role in protein quality control. Intriguingly, its ER-membrane counterpart, CNX, did not emerge in our screen, despite the inclusion of six sgRNA targeting CNX in the CRISPR library. This suggests a special role of CRT in the regulation of ATF6α, at least in CHO-K1 cells. CRT may not be the only repressor of ATF6α;

this role may be shared by BiP, which has been widely suggested to suppress all the three branches of the UPR under basal conditions. Nevertheless, because of BiP's pervasive effects on ER proteostasis and its role in direct repression of the IRE1 branch, its role in regulating ATF6 might have been missed by this screen.

Disruption of CRT in the *XC45-6S* dual ATF6α/IRE1 UPR reporter cell line selectively activated the ATF6α pathway reporter under basal conditions. This activation was accompanied by an increased baseline trafficking of ATF6α from the ER to the Golgi, enhanced processing of ATF6α to its active form and increased endogenous BiP protein levels. Loss of CRT also resulted in a conspicuous basal downregulation of the XBP1s::mCherry reporter that was associated with a reduced activity of IRE1 in *CRTΔ* cells. These findings align with previous evidence that N-ATF6α suppresses IRE1 signalling (*Walter et al., 2018*), and highlights the importance of ATF6α/IRE1 crosstalk.

CRT engages monoglycosylated glycoproteins in the ER via its glycan-binding lectin domain, retaining them in the CRT/CNX cycle (*Hebert et al., 1996*; *Ou et al., 1993*). A role for CRT's interaction with ATF6α_LD glycans in repressing ATF6α is suggested by the markedly diminished ability of a lectin mutant CRT[Y92A] to repress the ATF6α pathway compared to the WT. However, it is worth noting that CRT has been observed to bind proteins also in a glycan-independent manner (*Wijeyesakere et al., 2013*). Both BLI data and immunoprecipitation from cell extracts revealed that CRT can associate with both WT and unglycosylated ATF6α (ΔGly). In the former, direct interaction is characterised by slow kinetics, consistent with previous measurements of glycosylation-independent binding of CRT to its clients (Table 1 from *Wijeyesakere et al., 2013*). These biophysical features, coupled with the residual ability of the lectin mutant CRT[Y92A] to repress endogenous ATF6α, lead us to suggest that CRT repression of ATF6α may involve both glycosylation-dependent and -independent interactions.

Currently, the structural basis for ATF6α_LD interactions with CRT remains unknown, and its role in the direct or indirect retention of ATF6α in the ER is a subject for speculation. Nonetheless, the

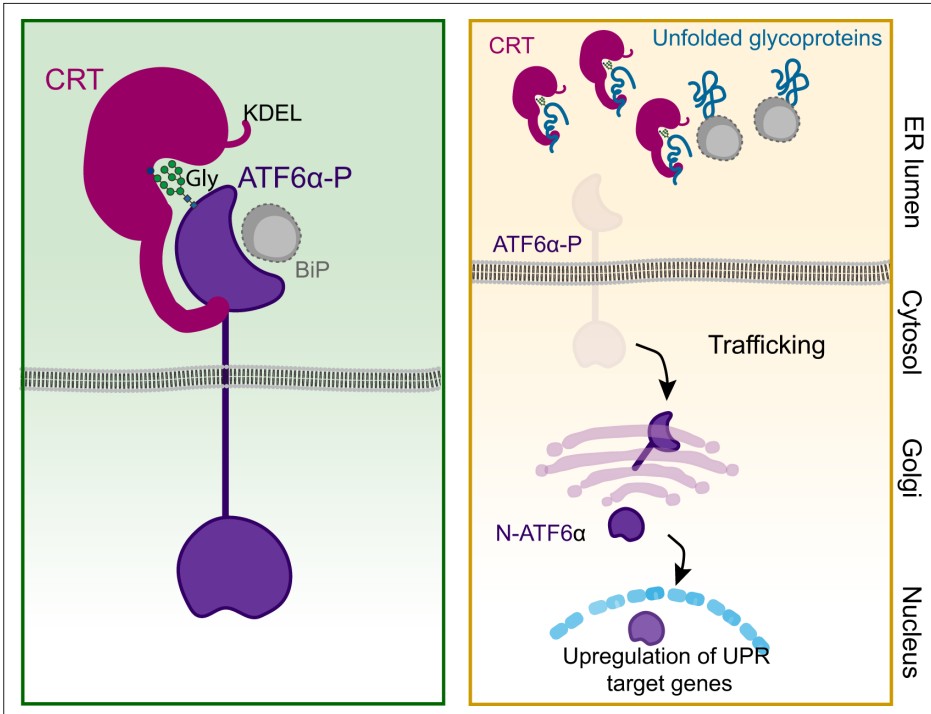

**Figure 8.** Proposed regulatory model of ATF6α by calreticulin (CRT) proteostasis. *Left panel*: When concentrations of competing ligands (unfolded glycoproteins) are low, endoplasmic reticulum (ER) retention of the ATF6α–CRT complex is favoured (via CRT's KDEL signal). The ATF6α–CRT interaction is cartooned here as having both glycan (Gly)-dependent and -independent components. *Right panel*: Unfolded glycoproteins compete with ATF6α for CRT and their increasing concentration favours ATF6α trafficking to the Golgi in stressed cells. A similar mechanism, involving BiP and shared by other unfolded protein response (UPR) transducers, has been previously proposed (*Bertolotti et al., 2000*) and is included here as a complement to ATF6α-specific regulation by CRT. ATF6-P refers to the precursor form of ATF6α, while N-ATF6α denotes the active form of ATF6α.

observations presented here support previous studies indicating that the stable interaction of CRT with cellular substrates combines both lectin-dependent and -independent interactions into a hybrid mode. Moreover, our observations raise the possibility that the lectin activity of CRT is dispensable for its interaction with ATF6α_LD but implicated in recruitment of a third component required for repression. Either way titration of CRT by competing ligands is a plausible regulatory mechanism coupling ATF6α activity to the burden of unfolded (glycol) proteins in the ER (*Figure 8*).

# Materials and methods

## Key resources table

| Reagent type (species) or resource | Designation | Source or reference | Identifiers | Additional information |
|---|---|---|---|---|
| Recombinant DNA reagent | pRK793 His6_TEV(S219V)_Arg | Gift from Wang Lab | UK759 | His-TEV(S219V)-Arg from pRK793 |
| Recombinant DNA reagent | pSpCas9(BB)-2A-Puro | Gift from Richard Timms | UK1367 | Puro-tagged CRISPR introduces double strand breaks (Addgene 48139) (Zhang PX459) |
| Recombinant DNA reagent | Lenti-Cas9-Blast | *Harding et al., 2019* | UK1674 | Lenti-Cas9-blasticidin resistance to make stable Cas9 expressing cells for CRISPR KO screening |
| Recombinant DNA reagent | pMD2.G | Gift from Didier Trono | UK1700; RRID:Addgene_12259 | Addgene plasmid #12259, lentiviral packaging helper (VSVG) |
| Recombinant DNA reagent | psPAX2 | Gift from Didier Trono | UK1701; RRID:Addbebe_12260 | Addgene plasmid #12260, lentiviral packaging helper |
| Recombinant DNA reagent | pKLV-U6gRNA(BbsI)PGKpuro2ABFP | *Koike-Yusa et al., 2014* | UK1789; RRID:Addgene_73542 | Addgene #50946, BFP-2A-Puro-tagged gRNA vector |
| Recombinant DNA reagent | pKLV-CHO_libA-PGKpuro2ABFP | *Ordóñez et al., 2021* | UK2561 | CRISPR-KO library of 125,030 selected guides for whole CHO genome screening |
| Recombinant DNA reagent | cgROSA26_BiP::sfGFP_HDR | This study | UK2846 | DNA repair plasmid to integrate an ATF6 reporter (BiP::sfGFP) at ROSA26 locus in CHO genome |
| Recombinant DNA reagent | cgROSA26_sgRNA_pSpCas9(BB)-2A-Puro | This study | UK2847 | Puro-tagged sgRNA/Cas9 introduce double strand breaks in cgROSA26 |
| Recombinant DNA reagent | pSpCas9 (BB)-2A-mTurquoise | This study | UK2915 | pSpCas9(BB)-2A vector to express mTurquoise together with guide RNA and Cas9 |
| Recombinant DNA reagent | cgATF6α_sgRNA_Ex8_pSpCas9(BB)-2A-mTurquoise | This study | UK2919 | mTurquoise-tagged sgRNA/Cas9 introduces double strand breaks in exon 8 of cgATF6α |
| Recombinant DNA reagent | cgERp18_sgRNA_g1_pSpCas9(BB)-2A-mTurquoise | This study | UK2920 | mTurquoise-tagged sgRNA/Cas9 introduces double strand breaks in exon 2 of TXNDC12 |
| Recombinant DNA reagent | cgERp18_sgRNA_g2_pSpCas9(BB)-2A-mTurquoise | This study | UK2921 | mTurquoise-tagged sgRNA/Cas9 introduces double strand breaks in exon 5 of TXNDC12 |
| Recombinant DNA reagent | cgIRE1sgRNA_lentiGuide-Puro | This study | UK 2929 | Lentiviral vector expressing cgIRE1 CRISPR guide targeting exon 18 of IRE1 without expression of Cas9 |
| Recombinant DNA reagent | cgATF6α_sgRNA_Ex2_pSpCas9(BB)-2A-mCherry (MP1) | This study | UK2942 | mCherry-tagged sgRNA/Cas9 introduces double strand breaks in exon 2 of cgATF6α |
| Recombinant DNA reagent | cgATF6_3xFLAG-mGL-TEV_HDR_pBS_V3 (MP1) | This study | UK2955 | HDR template for mGreenLantern knock-in at N-term of Hamster ATF6 |
| Recombinant DNA reagent | SP_FLAGM1_9E10_BirA_WT_KDEL_pCDNA5_FRT_TO_MP3 | This study | UK2969 | WT BirA, targeted to ER (with a signal peptide and a KDEL) |
| Recombinant DNA reagent | cgFURIN_sgRNA_Ex4_pSpCas9(BB)-2A-mTurquoise | This study | UK2988 | mTurquoise-tagged sgRNA/Cas9 introduces double strand breaks in exon 4 of cgFURIN |
| Recombinant DNA reagent | cgFURIN_sgRNA_Ex10_pSpCas9(BB)-2A-mTurquoise | This study | UK2989 | mTurquoise-tagged sgRNA/Cas9 introduces double strand breaks in exon 10 of cgFURIN |
| Recombinant DNA reagent | cgCRT_sgRNA_Ex1_pSpCas9(BB)-2A-mTurquoise | This study | UK2990 | mTurquoise-tagged sgRNA/Cas9 introduces double strand breaks in exon 1 of cgCRT |
| Recombinant DNA reagent | cgCRT_sgRNA_Ex3_pSpCas9(BB)-2A-mTurquoise | This study | UK2991 | mTurquoise-tagged sgRNA/Cas9 introduces double strand breaks in exon 3 of cgCRT |
| Recombinant DNA reagent | cgATF6_g1A_ex9_pSpCas9(BB)2A-mTurquoise_MP2 | This study | UK3021 | mTurquoise-tagged sgRNA/Cas9 introduces double strand breaks in exon 9 of cgATF6α (PR3125 and 3126 based) |
| Recombinant DNA reagent | cgATF6_g2A_ex10_pSpCas9(BB)2A-mTurquoise_MP2 | This study | UK3024 | mTurquoise-tagged sgRNA/Cas9 introduces double strand breaks in exon 10 of cgATF6α (PR3133 and 3134 based) |

*Continued on next page*

*Continued*

| Reagent type (species) or resource | Designation | Source or reference | Identifiers | Additional information |
|---|---|---|---|---|
| Recombinant DNA reagent | cgATF6_g2B_ex10_pSp Cas9(BB)2A-mTurquoise_MP2 | This study | UK3025 | mTurquoise-tagged sgRNA/Cas9 introduces double strand breaks in exon 10 of cgATF6α (PR3135 and 3136 based) |
| Recombinant DNA reagent | cgATF6_TV_EX8-9_Strep-Tag-Twin_V.1.3_pUC57 | This study | UK3054 | Genescript plasmid encodes cgATF6 LD$^{WT}$ for rescue experiments |
| Recombinant DNA reagent | cgATF6_410-659_noGly_V1_pUC57 | This study | UK3086 | pUC57 with ATF6 CDS gene fragment (all three glycans removed) for rescue experiments |
| Recombinant DNA reagent | EGFP_cgATF6_TEV_410-659_ S12MUT_AviTag_pCEFL_pu (MP4) | This study | UK3122 | EGFP replacing the cytosolic domain of cgATF6 LD with a -AviTag and with S1P/S2P mutations in |
| Recombinant DNA reagent | EGFP_cgATF6_TEV_410-659_ S12MUT_noGly_AviTag_pCEFL_pu (MP1) | This study | UK3141 | Glycan-free counterpart of UK3122 (TEV cleavage after TM) |
| Recombinant DNA reagent | rCRT_18-416_pSUMO3 (MP3) | This study | UK3142 | Bacterial expression of luminal fragment of rat Calreticulin |
| Recombinant DNA reagent | cgATF6_TV_EX8-9_Strep-tagtwin_CtoS_pUC57 mp7 | This study | UK3147 | Recombination template, UK3054 based, two cysteines changed to serine for rescue experiments |
| Recombinant DNA reagent | EGFP_cgATF6_TEV_410-659_S1& 2MUT_CtoS_AviTag_pCEFL_pu_MP2 | This study | UK3157 | Cysteine free counterpart of UK3122 (TEV cleavage after TM) |
| Recombinant DNA reagent | cgCNX_sgRNA_Ex2_pSp Cas9(BB)-2A-mTurquoise_MP1 | This study | UK3206 | mTurquoise-tagged sgRNA/Cas9 introduces double strand breaks in exon 2 of cgCNX (PR3364 and 3365 based) |
| Recombinant DNA reagent | cgCNX_sgRNA_Ex4_pSp Cas9(BB)-2A-mTurquoise_MP2 | This study | UK3207 | mTurquoise-tagged sgRNA/Cas9 introduces double strand breaks in exon 4 of cgATF6α (PR3366 and 3367 based) |
| Recombinant DNA reagent | hCRT_WT_pCEFL_BFP | This study | UK3231 | BFP-marked mammalian expression of untagged, WT human Calreticulin |
| Recombinant DNA reagent | hCRT_Y92A_pCEFL_BFP | This study | UK3232 | BFP-marked mammalian expression of untagged, human Calreticulin lectin mutant |
| Recombinant DNA reagent | hCRT_Y92A&W244A_pCEFL_BFP | This study | UK3233 | BFP-marked mammalian expression of untagged, human Calreticulin functional doble mutant |
| Sequence-based reagent | cgATF6a_ex2_sgRNA_1S | This study | 3010 | CACCGGCTGGTGCGGCCGGGACCA |
| Sequence-based reagent | cgATF6a_ex2_sgRNA_2AS | This study | 3011 | AAACTGGTCCCGGCCGCACCAGCC |
| Sequence-based reagent | cgATF6a_sgRNA_ex8_1S | This study | 2921 | CACCGAAGAAGAAGGAGTATATGC |
| Sequence-based reagent | cgATF6a_sgRNA_ex8_1S | This study | 2922 | AAACGCATATACTCCTTCTTCTTC |
| Sequence-based reagent | cgATF6a_sgRNA_ex9_1S | This study | 3125 | CACCGTTTCCTAGATTGCTGTGCTA |
| Sequence-based reagent | cgATF6a_sgRNA_ex9_2AS | This study | 3126 | AAACTAGCACAGCAATCTAGGAAAC |
| Sequence-based reagent | cgATF6a_sgRNA_ex10_1S | This study | 3133 | CACCGCATTTATAATGCTGAACTA |
| Sequence-based reagent | cgATF6a_sgRNA_ex10_2AS | This study | 3134 | AAACTAGTTCAGCATTATAAATGC |
| Sequence-based reagent | cgATF6a_sgRNA_ex10_3S | This study | 3135 | CACCGAGGTGAGTATTCAAGTGTT |
| Sequence-based reagent | cgATF6a_sgRNA_ex10_4AS | This study | 3136 | AAACAACACTTGAATACTCACCTC |
| Sequence-based reagent | cgCRT_sgRNA_ex1_1S | This study | 3068 | CACCGTTCCGCGGCGGCCAGGCCG |
| Sequence-based reagent | cgCRT_sgRNA_ex1_2AS | This study | 3069 | AAACCGGCCTGGCCGCCGCGGAAC |
| Sequence-based reagent | cgCRT_sgRNA_ex3_1S | This study | 3070 | CACCGCGCGTAAAATCGGGCATCC |
| Sequence-based reagent | cgCRT_sgRNA_ex3_2AS | This study | 3071 | AAACGGATGCCCGATTTTACGCGC |
| Sequence-based reagent | cgFURIN_sgRNA_ex4_1S | This study | 3064 | CACCGTGGTCTCCATCCTGGACGA |

*Continued on next page*

*Continued*

| Reagent type (species) or resource | Designation | Source or reference | Identifiers | Additional information |
|---|---|---|---|---|
| Sequence-based reagent | cgFURIN_sgRNA_ex4_2AS | This study | 3065 | AAACTCGTCCAGGATGGAGACCAC |
| Sequence-based reagent | cgFURIN_sgRNA_ex10_1S | This study | 3066 | CACCGCCACCTCAATGCTAATGAT |
| Sequence-based reagent | cgFURIN_sgRNA_ex10_2AS | This study | 3067 | AAACATCATTAGCATTGAGGTGGC |
| Sequence-based reagent | cgCNX_sgRNA_ex2_1S | This study | 3364 | CACCGCAAAGCTCCAGTTCCAACAG |
| Sequence-based reagent | cgCNX_sgRNA_ex2_2AS | This study | 3365 | AAACCTGTTGGAACTGGAGCTTTGC |
| Sequence-based reagent | cgCNX_sgRNA_ex4_3S | This study | 3366 | CACCGGCTTGGTATCAAACAGGAA |
| Sequence-based reagent | cgCNX_sgRNA_ex4_4AS | This study | 3367 | AAACTTCCTGTTTGATACCAAGCC |
| Sequence-based reagent | cgERp18_sgRNAg1_ex2_1S | This study | 2962 | CACCGTTTTGGAGATCATATTCAC |
| Sequence-based reagent | cgERp18_sgRNAg1_ex2_2AS | This study | 2963 | AAACGTGAATATGATCTCCAAAAC |
| Sequence-based reagent | cgERp18_sgRNAg2_ex5_1S | This study | 2964 | CACCGTGGAATATAACCCCCATCA |
| Sequence-based reagent | cgERp18_sgRNAg2_ex5_2AS | This study | 2965 | AAACTGATGGGGGTTATATTCCAC |
| Sequence-based reagent | cgXBP1.19S5 | This study | 1470 | GGCCTTGTAATTGAGAACCAGGAG |
| Sequence-based reagent | cgXBP1.14AS | This study | 5 | GAATGCCCAAAAGGATATCAGACTC |
| Antibody | Polyclonal Rabbit anti-calreticulin | Abcam | Cat# ab19261, RRID:AB_955722 | Used at 1:1000 |
| Antibody | Polyclonal Chicken anti-calreticulin | Invitrogen | Cat# PA1-902A, RRID:AB_2069607 | Used at 1:1000 |
| Antibody | Polyclonal Rabbit anti-calnexin | Enzo Life Sciences | Cat# ADI-SPA-860-D, RRID:AB_2038898 | Used at 1:1000 |
| antibody | Polyclonal Rabbit anti-FURIN | Abcam | Cat# ab3467, RRID:AB_303828 | Used at 1:1000 |
| Antibody | Monoclonal Mouse anti IRE1a_LD serum | *Bertolotti et al., 2000* | NY200 | Used at 1:1000 |
| Antibody | Monoclonal Rabbit anti p-IRE1 | Genetech | | Used at 1:1000 |
| Antibody | Polyclonal Chicken anti-hamster BiP | *Avezov et al., 2013* | anti-BiP | Used at 1:1000 |
| Antibody | Monoclonal Mouse anti FLAG-M2 | Sigma | Cat# F1804, RRID:AB_262044 | Used at 1:1000 |
| Antibody | Polyclonal Rabbit anti GFP | *Marciniak et al., 2004* | NY | Used at 1:1000 |
| Antibody | Monoclonal Mouse anti KDEL (10C3) | Enzo Life Sciences | ENZ-ABS679 | Used at 1:1000 |
| Antibody | Monoclonal Mouse anti-actin | Fisher Scientific | clone C4-691002 | Used at 1:1000 |
| Antibody | Monoclonal Rabbit anti-GAPDH | Cell Signalling | 14C10; mab# 2118 | Used at 1:2000 |
| Antibody | Polyclonal rabbit polyclones anti-$\alpha$/$\beta$-Tubulin | Cell Signalling | CST2148S | Used at 1:1000 |
| Chemical compound, drug | DMEM | Sigma | D6429 | |
| Chemical compound, drug | HyClone II Serum | Thermo Fisher Scientific | SH30066.03 | |
| Chemical compound, drug | Penicillin/Streptomycin | Sigma | P0781 | |
| Chemical compound, drug | L-Glutamine | Sigma | G7513 | |
| Chemical compound, drug | Non-essential amino acids solution | Sigma | M7145 | |

*Continued on next page*

*Continued*

| Reagent type (species) or resource | Designation | Source or reference | Identifiers | Additional information |
|---|---|---|---|---|
| Chemical compound, drug | Nutrient Mixture F12 | Sigma | N4888 | |
| Chemical compound, drug | Lipofectamine LTX | Thermo Fisher Scientific | A12621 | |
| Chemical compound, drug | Puromycin | MERCK-milipore | 540222 | |
| Chemical compound, drug | EDTA-free Protease inhibitor Cocktail | Roche | 11873580001 | |
| Chemical compound, drug | Anti-FLAG M2 Affinity Gel | Sigma | F3165 | |
| Chemical compound, drug | ChromoTek GFP-Trap Agarose | Thermo Fisher Scientific | Tag-20 | |
| Chemical compound, drug | Tunicamycin | Melford | T2250 | |
| Chemical compound, drug | 2-Deoxyglucose (2DG) | ACROS Organics | D6134 | |
| Chemical compound, drug | Ceapin A7 | Sigma | SML2330 | |
| Chemical compound, drug | Thapsigargin | Calbiochem | CAS 67526-95-8 | |
| Chemical compound, drug | PF-429242 dihydrochloride (S1P inhibitor) | Tocris | SML0667 | |
| Chemical compound, drug | 4µ8C | Tocris Bioscience | 4479 | |
| Chemical compound, drug | TRIzolä Reagent | Ambion/Invitrogen | 15596026 | |
| Chemical compound, drug | PureLinkä RNA mini kit | Invitrogen | 12183018A | |
| Chemical compound, drug | RevertAid Reverse Transcriptase | Thermo Scientific | EP0441 | |
| Chemical compound, drug | FURIN | New England Biolabs | P8077S | |
| Chemical compound, drug | MBP5-FN-paramyosin-ΔSal | New England Biolabs | E8052S | |
| Chemical compound, drug | MycoAlert (TM) Mycoplasma Detection Kit | Lonza | LT07-118 | |
| Cell line (*Cricetulus griseus*) | CHO XC45-6S | This study | XC45-6S | Parental ATF6α/IRE dual UPR reporter CHO cell line expressing XBP1s::mCherry and BiP:sfGFP reporters |
| Cell line (*Cricetulus griseus*) | XC45-6S cells-Cas9 stable#21 | This study | XC45-6S-Cas9 stable#21 | Parental ATF6α/IRE dual UPR reporter CHO cell line expressing XBP1s::mCherry and BiP:sfGFP reporters stably expressing Cas9 |
| Cell line (*Cricetulus griseus*) | CHO-K1 *XC45-6S*, XBP1s::mCherry & BiP:sfGFP *ATF6α*Δ (KO) | This study | XC45-6S, *ATF6*Δ, *#B6* | ATF6α/IRE dual reporter cell line with depletion of ATF6_clone B6 |
| Cell line (*Cricetulus griseus*) | CHO-K1 *XC45-6S*, XBP1s::mCherry & BiP:sfGFP *ATF6α*Δ (KO) | This study | XC45-6S, *ATF6*Δ, *#5A2* | ATF6α/IRE dual reporter cell line with depletion of ATF6_clone 5A2 |
| Cell line (*Cricetulus griseus*) | CHO-K1 *XC45-6S*, ATF6αLD$^{WT}$ rescue | This study | XC45-6S, *ATF6α*LD$^{WT}$ *#R2_9* | ATF6α/IRE dual reporter cell line with depletion of ATF6_LD and rescue with *ATF6α*LD$^{WT}$ _clone R2.9 |
| Cell line (*Cricetulus griseus*) | CHO-K1 *XC45-6S*, ATF6αLD$^{ΔGly}$ rescue | This study | XC45-6S, *ATF6α*LD$^{ΔGly}$*#GF6* | ATF6α/IRE dual reporter cell line with depletion of ATF6_LD and rescue with *ATF6α*LD$^{ΔGly}$_clone 6 |
| Cell line (*Cricetulus griseus*) | CHO-K1 *XC45-6S*, ATF6αLD$^{ΔC}$ rescue | This study | XC45-6S, *ATF6α*LD$^{ΔC}$*#CS10* | ATF6α/IRE dual reporter cell line with depletion of ATF6_LD and rescue with *ATF6α*LD$^{ΔC}$_clone 10 |
| Cell line (*Cricetulus griseus*) | CHO-K1 3xFLAG-mGL_ATF6α knock-in cells | This study | 2K cells | CHO-K1 cells with endogenous ATF6 tagged with a 3xFLAG-mGL tag |

*Continued on next page*

*Continued*

| Reagent type (species) or resource | Designation | Source or reference | Identifiers | Additional information |
|---|---|---|---|---|
| Cell line (*Cricetulus griseus*) | CHO-K1 GFP-cgATF6αLD_S1P&S2P$^{mut}$ | This study | UK3122 cells | CHO-K1 stable expressing a GFP-cgATF6αLD_S1P&S2P$^{mut}$ |
| Cell line (*Cricetulus griseus*) | CHO-K1 *XC45-6S, CRTΔ#2P* | This study | XC45-6S, CRTαΔ#2P | ATF6α/IRE dual reporter cell line with depletion of CRT_clone 2P |
| Cell line (*Cricetulus griseus*) | CHO-K1 *XC45-6S, CRTΔ#2Q* | This study | XC45-6S, CRTαΔ#2Q | ATF6α/IRE dual reporter cell line with depletion of ATF6-clone 2Q |
| Cell line (*Cricetulus griseus*) | CHO-K1 *XC45-6S, FURINΔ#2.2.G* | This study | XC45-6S, FURINΔ#2.2.G | ATF6α/IRE dual reporter cell line with depletion of Furin_ clone 2.2.G |
| Cell line (*Cricetulus griseus*) | CHO-K1 *XC45-6S, CNXΔ#6* | This study | XC45-6S, CNXΔ#6 | ATF6α/IRE dual reporter cell line with depletion of CNX_clone 6 |
| Cell line (*Cricetulus griseus*) | CHO-K1 *XC45-6S, ERp18Δ#7E2* | This study | XC45-6S, ERp18Δ#7E2 | ATF6α/IRE dual reporter cell line with depletion of ERp18_clone 7E2 |
| Software, algorithm | MAGeCK | *Li et al., 2014* | | |
| Software, algorithm | Metascape | *Zhou et al., 2019* | | |
| Software, algorithm | MacVector | | | |
| Software, algorithm | Photoshop and Illustrator | | | |
| Software, algorithm | FlowJo | | | |
| Software, algorithm | ImageJ | | | |
| Software, algorithm | GraphPad-Prism V8 | | | |
| Software, algorithm | Volocity V6.3 | PerkinElmer | | |

## Mammalian cell culture, transfections, and treatments

Adherent Chinese Hamster Ovary (CHO-K1) cells (ATCC CCL-61) were maintained in Ham's nutrient mixture F12 (Sigma) supplemented with 10% (vol/vol) serum (FetalClone-2, HyClone), 2 mM l-gluta-mine (Sigma), and 1% penicillin/streptomycin (Sigma). HEK293T cells (ATCC CRL-3216) were cultured in Dulbecco's Modified Eagle Medium (DMEM) cell media (Sigma) supplemented as above. All cells were grown in a humidified 37°C incubator with 5% $CO_2$ and passed every 2–3 days. When indicated, cells were treated with 5 µM Ceapin A7 (SML2330; Sigma), 15 µM S1P inhibitor (PF429242 dihydrochloride; Tocris), 2.5 µg/ml Tunicamycin (Tm, Melford), 4 mM 2DG (ACROS Organics), 0.5 µM Thapsigargin (Tg; Calbiochem), and 16 µM 4µ8C (Sigma) for the indicated times. All compounds were diluted in pre-warmed culture medium and immediately added to the cells by medium exchange. Untreated cells were treated with dimethylsulfoxide (DMSO) solvent vehicle control. Transfections in CHO-K1 cells were performed using Lipofectamine LTX (Thermo Fisher Scientific, USA) at 1:3 DNA (µg) to LTX (µl) ratio, as for HEK293T using TransIT-293 Transfection Reagent (MIR2704, Mirus), according to the manufacturer's instructions. Typically, cells were seeded at a density of $2.5 \times 10^5$ cells/well in 6-well plates a day before transfection and analysed 48–72 hr after transfection.

The cell lines used in this study have tested negative for mycoplasma contamination using the MycoAlert Mycoplasma Detection Kit (Lonza). None of the cell lines appear on the list of commonly misidentified cell lines maintained by the International Cell Line Authentication Committee. The identity of the CHO-K1 cell lines has been authenticated by successfully targeting essential genes using a species-specific CRISPR whole-genome library and sequencing the WT or mutant alleles of the genes studied, which confirmed the sequences reported for the corresponding genome.

## Mammalian cell lysates and immunoblotting

CHO-K1 or HEK293T cells were cultured in 6-, 12-well plates, or 10-cm dishes until reaching 95% confluence. Cells were washed twice with prechilled phosphate-buffered saline (PBS) on ice, and whole-cell extracts were scrapped out in 1 mM EDTA (Ethylenediaminetetraacetic Acid)–PBS, pelleted at 370 × *g* for 10 min at 4°C and incubated in Nonidet P40 (NP-40) lysis buffer (150 mM NaCl, 50 mM Tris–HCl pH 7.5, 1% (vol/vol) NP-40) supplemented with 2× protease inhibitor cocktail (Roche Applied Science) for 30 min. Next, the samples were clarified at 21,130 × *g* for 20 min at 4°C and the supernatants were transferred to fresh tubes. Protein concentration was determined using Bio-Rad protein assay. To assess the interaction between CRT and ATF6α through either BLI or co-immunoprecipitation, the lysis

buffer was supplemented with 5 mM $CaCl_2$. The protein samples were separated on 8–12.5% SDS–PAGE under reducing conditions and transferred onto PVDF (Polyvinylidene Fluoride) membranes as described previously (*Ordóñez et al., 2013*). Membranes were probed with the following primary antibodies: rabbit polyclonal anti-calreticulin (1:1000) (ab19261, abcam; to detect human CRT); chicken polyclonal anti-calreticulin (1:1000) (PA1-902A, Invitrogen; to detect Chinese hamster CRT); rabbit polyclonal anti-calnexin (1:1000) (SPA-860, Enzo Life Sciences); rabbit polyclonal anti-FURIN (1:1000) (ab3467, abcam); mouse monoclonal anti IRE1_LD serum (NY200) (1:1000) *Bertolotti et al., 2000*; rabbit monoclonal anti p-IRE1 (1:1000) (Genetech); rabbit polyclonal anti GFP (NY1066) (1:1000) (*Marciniak et al., 2004*) made against purified bacterially expressed GFP following removal of the GST affinity tag via thrombin cleavage site; chicken anti-BiP IgY (1:1000) *Avezov et al., 2013*; mouse monoclonal anti-FLAG M2 (1:1000) (F1804, Sigma); mouse monoclonal anti-KDEL (10C3) (1:1000) (ENZ-ABS679; Enzo Life Sciences); mouse monoclonal anti-actin (1:1000) (clone C4-691002, Fisher Scientific); rabbit monoclonal anti-GAPDH (1:2000) (14C10; Cell Signalling Technology); rabbit polyclonal anti-α/β-Tubulin (1:1000) (CST2148S, Cell Signalling Technology). For secondary antibody, IRDye fluorescently labelled antibodies or horseradish peroxidase antibodies were used. Membranes were scanned using an Odyssey near infrared imager (LI-COR) and signals were quantified using ImageJ software.

Signal quantification and analysis were performed using Prism 10 (GraphPad).

## Flow cytometry and FACS

To analyse UPR reporter activities, cells were grown on 6- or 12-well plates until reaching 80-90% confluence and then treated as indicated. For flow cytometry analysis, cells were washed twice in PBS, collected in PBS supplemented with 4 mM EDTA, and 20,000 cells/sample were analysed by multi-channel flow cytometry on a LSRFortessa cell analyser (BD Biosciences). For FACS, cells were collected in PBS containing 4 mM EDTA and 0.5% bovine serum albumin (BSA) and then sorted on a Beckman Coulter MoFlo cell sorter. Sorted cells were either collected in fresh media as a bulk of cells or individually sorted into 96-well plates and then expanded. Gating for live cells was based on FSC-A/SSC-A and for singlets was based on FSC-W/SSC-A. BiP::sfGFP and mGreenLantern (mGL) fluorescent signals were detected with an excitation laser at 488 nm and a 530/30 nm emission filter; XBP1s::mCherry fluorescence with an excitation laser 561 nm and a 610/20 nm emission filter, while mTurquoise and BFP fluorescence with an excitation laser 405 nm and a 450/50 nm filter emission. Data analysis was performed using FlowJo V10, and median reporter analysis was conducted using Prism 10 (GraphPad).

## Lentiviral production

Lentiviral particles were produced by co-transfecting HEK293T cells with the library plasmid (UK2561), the packaging plasmids psPAX2 (UK1701) and pMD2.G (UK1700) at 10:7.5:5 ratio using TransIT-293 Transfection Reagent (MIR2704, Mirus) according to the manufacturer's instructions. Eighteen hours after transfection, medium was changed to medium supplemented with 1% BSA (Sigma). The supernatant containing the viral particles was collected 48 hr after transfection, filtering through a 0.45-µm filter, and directly used to infect CHO-K1 cells seeded in 6-well plates for viral titration to calculate the amount of virus to use aiming a low MOI around 0.3 to ensure a single integration even per cell.

## Generation of the *XC45-6S* double UPR reporter cell by genome editing in *ROSA26* locus of CHO-K1 cells

The putative *ROSA26* locus in CHO cells has been recently identified and described as a 'safeharbour' site for heterologous gene expression and stable long-term expression of a transgene (*Gaidukov et al., 2018*). To generate a double UPR reporter cell line, we used a previously described *XC45* CHO-K1 cell line bearing a XBP1s::mCherry reporter expressing a pCAX-F-XBP1ΔDBD-mCherry transgene randomly integrated in the genome (*Harding et al., 2019*), for targeting the *ROSA26* locus for knock-in with a landing pad cassette, comprised the *cg*BiP promoter region containing ERSE-I elements, fused to the superfolded (sf) GFP into the CHO genome by CRISPR/Cas9 integration to make this reporter cell line also sensitive to ATF6α activity. The landing pad donor vector was constructed by appending homology arm sequences of *ROSA26* locus (440 bp predefined sequence from Gaidukov's paper) to the 5′ promoter region of *cg*BiP (1000 bp), along with the first eight amino

acids of the BiP CDS fused to the sfGFP. Homology arms were PCR amplified from CHO genomic DNA using predefined primers from Gaidukov's paper. Subsequently, the *cg*BiP promoter, a gene-block containing sfGFP, and the flanking homology arms were fused by Gibson assembly cloning method, which allows multiple DNA fragments joining in a single, isothermal reaction (1 hr at 50°C). Targeted integration was performed by co-transfection with the circular landing pad donor vector (UK2846) and a single sgRNA/Cas9 targeting *cgROSA26* locus (UK2847). Transfected cells were selected with puromycin (8 µg/ml) for 3 days and then treated with 2DG (4 mM) for 20 hr to induce ER stress prior FACS. Cells were phenotypically sorted into 96-well plates by their activation of both UPR reporters (mCherry^high; GFP ^high). Phenotypically selected clones were subsequently confirmed by correct genetic integration. For stability integration, clones were maintained and expanded every 2–3 days for 1 month and weekly analysed with flow cytometry for mCherry and GFP expression under treatment with different ER stressors agents. For the genome-wide CRISPR/Cas9 screen, Cas9 was ultimately introduced by transduction with lentiviral particles prepared from packaging Lenti-Cas9 plasmid (UK1674). Cas9 activity was confirmed by targeting the exon 18 of IRE1 (*ERN1* locus) (UK2929) followed by induction of ER stress with 2DG (4 mM) for 20 hr (*Figure 1—figure supplement 1F*).

## CRISPR/Cas9 screen

The screen was performed following established protocols using a pooled CHO-K1 knockout CRISPR library containing 125,030 sgRNAs targeting 21,896 genes, with six guides per gene, as well as 1000 non-targeting sgRNAs as a negative control cloned into the lentiviral sgRNA expression vector pKLV-U6gRNA(BbsI)-PGKpuro2ABFP (*Shalem et al., 2014*; *Ordóñez et al., 2021*). Approximately $2.1 \times 10^8$ CHO-K1 Cas9 stable cells carrying the dual UPR reporter ATF6α/IRE1 (*XC45-6S* cells) were infected at an MOI of 0.3 to favour infection with a single viral particle per cell. Two days after infection, trans-duced cells were selected with 8 µg/ml puromycin for 14 days. Afterwards, transduced cells were split into two subpopulations: one was treated with 2DG to induce ER stress, while the other one remained untreated. For the *ATF6α activator screen*, conducted under ER stress conditions, less than 2% of cells that showed a XBP1s::mCherry^high; BiP::sfGFP^low phenotype were selected for analysis. Similarly, for the *ATF6α repressor screen*, conducted under basal (untreated) conditions, less than 2% of total sorted cells that showed a XBP1s::mCherry^low; BiP::sfGFP^high phenotype were selected for analysis. Rounds of enrichment were carried out through cellular recovery, expansion, and sorting. Equal number of transduced cells, both untreated or 2DG treated, was passed without sorting as a control group in each round of sorting. To prepare samples for deep sequencing, genomic DNA from enriched and sorted populations as well as unsorted cells (to represent the entire library) was extracted from ~$3.6 \times 10^7$ and ~$1–3 \times 10^6$ cells, respectively, by incubation in proteinase K solution [100 mM Tris–HCl pH 8.5, 5 mM EDTA, 200 mM NaCl, 0.25% (wt/vol) SDS, 0.2 mg/ml Proteinase K] overnight at 50°C. Integrated sgRNA sequences were amplified by nested PCR and the necessary adaptors for Illumina sequencing were introduced at the final round of amplification. The purified products were quantified and sequenced on an Illumina NovaSeq 6000 by 50 bp single-end sequencing. Downstream analyses to obtain sgRNA read counts, gene rankings, and statistics were performed with MAGeCK software (*Li et al., 2014*). GO analyses were performed using Metascape with default parameters.

## Knockout cells using CRISPR/Cas9 technology

sgRNAs targeting exon regions of *C. griseus ATF6α*, *CRT*, *CNX*, *FURIN*, and *ERp18* were designed using CRISPR-Design tools from *CCTop* (https://cctop.cos.uniheidelberg.de:8043/index.html) and *CRISPy-Cas9 target finder for CHO-K1* (http://staff.biosustain.dtu.dk/laeb/crispy/) databases. sgRNAs were then cloned into the pSpCas9(BB)-2A-mTurquoise (UK2915) or pSpCas9(BB)-2A-Puro plasmids (UK1367) as previously reported (*Ran et al., 2013*). The integrity of the constructs was confirmed by DNA sequencing. To generate the knockout cells, 0.75–1 µg of sgRNA/Cas9 plasmids was transfected into *XC45-6S* cells using Lipofectamine LTX (Thermo Fisher) in a 6-well plate format. After 48 hr post-transfection, mTurquoise^high cells were sorted into 96-well plates at 1 cell/well using a MoFlo Cell Sorter (Beckman Coulter). For cells transfected with sgRNA/Puro plasmids, selection was performed in the presence of 8 µg/ml puromycin, and serial dilution was used to isolate single clones. The knock-outs cells were confirmed by Sanger sequencing and immunoblotting.

## ATF6α-tagged cells by CRISPR/Cas9 genome editing

To investigate potential modulators of ATF6α activity following the CRISPR/Cas9 high-throughput screen using biochemistry and microscopy-compatible tools, two mammalian cell lines were engineered. These cell lines either harboured a knock-in of the endogenous *cgATF6α* locus or carried a stable transgene encoding ATF6α_LD tagged with a bright GFP. The tags were positioned at the N-cytosolic terminus, known to be well tolerated.

### 3xFLAG-mGreenLantern_ATF6α knock-in cells (*2K* cells)

Protein alignment of *hs*ATF6α and *cg*ATF6α (UniProt database) indicated that translation initiation predominantly occurs at the second Met at the beginning of exon 2 (<u>ME</u>SP). Instead of tagging at the first Met in the CHO genome, tags were introduced before the third Met. CHO-K1 plain cells were offered with a unique pSpCas9(BB)-2A-mCherry targeting exon 2 of *cg*ATF6α locus (UK2942) and a donor plasmid containing homology arms flanking a minigene encoding a 3xFLAGmGreenLantern(mGL)-TEV sequence geneblock (UK2955) to create an endogenous tagged ATF6. After 48 hr post-transfection, mGL$^{high}$ cells were sorted as single cells into 96-well plates. Clones that preserved stable mGL fluorescent signal by flow cytometry and showed expected ATF6α processing upon ER stress induction (DTT) were subsequently confirmed by correct genetic integration. Microscopy characterisation was performed under basal conditions or after treatments with ER stressors (DTT), Ceapin A7 and S1P inhibitor. A clone referred to as *2K* was expanded and selected for functional analysis as a 3xFLAG-mGL-tagged endogenous ATF6α.

### GFP-cgATF6α_LD$^{S1P,S2Pmut}$ (*UK3122* cells)

ATF6 has a half-life of about 3 hr (**George et al., 2020**). To increase the proportion of intact, active N-ATF6α molecules reaching the Golgi, a genetic construct featured the TM and LD of ATF6α deliberately lacking S1P and S2P cleavage sites, and further strategically appended with an Avi-tag at the C-terminal, was engineered. This construct was then used for a dual purpose: as a tool for both microscopy studies as well as to be biotinylated and purified to be used as a ligand in BLI experiments. Shen and Ye's research provided the basis for the design of S2P and S1P mutations (N391F and P394L for S2P mutant, and LL408/9VV for S1P mutant) (**Shen et al., 2002**; **Ye et al., 2000**). A DNA-plasmid encoding eGFP_cgATF6α_TEV_LD410-659_S12PMUT_AviTag_pCEFL_pu (UK3122) was introduced into CHO-K1 plain cells via Lipofectamine LTX. Transfected cells were selected for resistance to puromycin (8 μg/ml), and the isogenic clones were selected for confocal microscopy assays and *in vitro* BLI interaction assays.

## A genetic platform to generate cgATF6α LD variants knock-in cells

### ATF6αΔ

Two sgRNA sequences targeting exons 9 and 10 of cg*ATF6α* were incorporated into pSpCas9(BB)-2A-mTurquoise plasmid (UK2915) to generate two sgRNA/Cas9 expression plasmids (UK3021 and UK3024) according to the published procedure (**Ran et al., 2013**). These plasmids were then transfected into CHO-K1 *XC45-6S* double UPR reporter cells. After 72 hr, mTurquoise$^{high}$ cells were sorted as single cells into 96-well plates. Genomic DNA from clones showing no response to ATF6α activation (BiP::sfGFP$^{low}$) upon tunicamycin treatment (0.5 μM, 20 hr) was extracted for Sanger sequencing. Clones with frameshifts causing deletions in both alleles of the DNA region flanking by both sgRNAs were selected for further functional assays.

### cgATF6αLD 'minigenes' knock-in

A unique sgRNA sequence targeting the exon 10 of α*ATF6α*Δ cells was inserted into the pSpCas9(BB)-2A-mTurquoise plasmid to create the sgRNA/Cas9 plasmid UK3025. Three repair 'minigenes' templates were constructed: A WT, glycan-free (ΔGly), or a cysteine-free (ΔC) LD to be reconstituted in ATF6αΔ cells by CRISPR/Cas9-mediated HDR.

For the construction of the WT LD repair template (WT; UK3054), a minigene block that contains exons 10–17 of *cgATF6α* (785 bp; GenScript) was digested with *Age*I and *Afl*II and ligated into *Age*I/*Afl*II-digested pUC57 plasmid. To create a Glycan free LD (ΔGly; UK3086) template, a minigene block containing exons 10–17 (GenScrip) with the three *N*-glycosylation sites were mutated by replacing

Thr 464, 575, and 634 with Ala, was cloned into pUC57 plasmid as previously described. Similarly, to generate a cysteine-free LD (ΔC; UK3147) template, a minigene block containing exons 10–17 (GenScrip) with Cys457 and Cys607 replaced with Ser, was cloned into pUC57 plasmid as previously performed. The integrity of the constructs was confirmed by DNA sequencing.

## Immunoprecipitation using GFP-Trap Agarose

Equal volumes of the cleared and normalised lysates were incubated with 15–20 µl GFP-Trap Agarose (ChromoTek) pre-equilibrated in lysis buffer. The mixture was incubated overnight at 4°C with rotation. The beads were then recovered by centrifugation (845 × $g$, 3 min) and washed four times with washing buffer (50 mM Tris–HCl, 1 mM EDTA, 0.1% Triton X-100) and one more with PBS. Proteins were eluted in 35 µl of 2× SDS–DTT sample for 5 min at 95°C. Equal sample volumes were analysed by SDS–PAGE and immunoblotting, as described above. Normalised cell lysates (15–30 µg) were loaded as an 'input' control.

## BLI, protein expression, and *in vivo* biotinylation

BLI studies were undertaken using on the Octet RED96 System (Pall FortéBio) using ATF6α_LD variants as biotinylated ligands that were immobilised on streptavidin (SA)-coated biosensors (Pall FortéBio) and rat calreticulin (*r*CRT) as the analyte. Prior to use, SA-coated biosensors were hydrated in a BLI buffer (150 mM NaCl, 20 mM Tris, 1 mM CaCl$_2$) for 10 min. BLI reactions were prepared in 200 µl volumes in 96-well microplates (Greiner Bio-One), at 30°C. Ligand loading was performed for 600 s until a binding signal of 1 nm displacement was reached. Following ligand immobilisation, the sensor was baselined in the reaction buffer for at least 200 s, after which it was immersed in wells containing the analyte for association for 1200s. After each association, the sensor was dipped into a buffer containing well to allow dissociation for 2400s.

Biotinylated ATF6αLD constructs [ATF6αLD$^{WT}$ (UK3122); ATF6αLD$^{ΔGly}$ (UK3141); and ATF6LD$^{ΔC}$ (UK3157)] were expressed in HEK293T cells, each one encoding a N-terminal GFP tag followed by a TEV protease cleavage site just after the TM and an Avi-tag at C-terminal. S1P and S2P cleavage sites were also mutated to increase expression yield. Biotinylation was conducted *in vivo*, where cells were co-transfected with a BirA-encoding plasmid (biotin ligase, UK2969) in a 1:0.06 ratio. After 24 hr, media was replaced with fresh media containing 50 µM biotin dissolved in DMSO and cells were harvested and lysed the following day. The lysis buffer was supplemented with 5 mM CaCl$_2$. Post immunoprecipitation using GFP-Trap Agarose, proteins were eluted by incubation with an excess of TEV protease (UK759, 16 µg) in 400 µl TEV cleavage buffer [(25 mM Tris–HCl pH 7.4, 150 mM NaCl, 0.05% NP-40, 0.05 mM CaCl$_2$, 0.05 mM tris(2-carboxyethyl)phosphine (TCEP))] for 16 hr at 4°C with orbital rotation. Cleavage products were snap-frozen in liquid nitrogen and stored at −80°C for BLI experiments.

WT full-length *r*CRT (UK3142) was expressed in *E. coli* cells (BL21 C3013). Bacterial cultures were grown in LB medium with 100 µg/ml ampicillin at 37°C to an OD$_{600\ nm}$ of 0.6 to 0.8. Expression was induced with 1 mM isopropyl β- d-1-thiogalactopyranoside (IPTG), and the cells were further grown for 16 hr at 18°C. After cell sedimentation by centrifugation, the pellets were lysed with a high-pressure homogeniser (EmulsiFlex-C3; Avestin) in bacterial lysis buffer (50 mM Tris–HCl pH 7.4, 500 mM NaCl, 2 mM CaCl$_2$, 20 mM imidzole) containing protease inhibitors (1 mM phenylmethylsulfonyl fluoride (PMSF), 2 µg/ml pepstatin, 8 µg/ml aprotinin, 2 µM leupeptin) and 0.1 mg/ml DNaseI. Typical yields from 2 l of bacterial culture were obtained.

## Immunoprecipitation using GFP-Trap Agarose

To investigate the interaction between CRT and ATF6α in cells, HEK293T cells were cultured in 10 cm dishes (3–4 plates per sample) and transfected with constructs expressing either GFPATF6α_LD$^{WT}$ (UK3122) or GFP-ATF6α_LD$^{ΔGly}$ (UK3141). Equal volumes of the cleared and normalised lysates were incubated with 15–20 µl GFP-Trap Agarose (ChromoTek) preequilibrated in lysis buffer. The mixture was incubated overnight at 4°C with rotation. The beads were then recovered by centrifugation (845 × $g$, 3 min) and washed four times with washing buffer (50 mM Tris–HCl, 1 mM EDTA, 0.1% Triton X-100). Proteins were eluted in 35 µl of 2× SDS–DTT sample for 5 min at 95°C. Equal sample volumes were analysed by SDSPAGE and immunoblotting, as described above. Normalised cell lysates (15–30 µg) were loaded as an 'input' control.

## Analysis of endogenous ATF6α processing in CRT-depleted cells

To investigate the impact of CRT depletion on ATF6α cleavage, CRT was depleted in *2K* cells 3xFLAG-mGL-ATF6α by CRISPR/Cas9 gene editing (UK2991). *2K-CRTΔ* derivative clones were confirmed by Sanger sequencing and selected for further experiments. Both *2K* parental cells and *2K-CRTΔ* derivative cells were cultured in 10-cm dishes for 2 days and then treated with 2DG (4 mM) for a time course up to 6 hr before harvested. Following treatment, the cells were lysed in Nonidet-lysis buffer for extracting the cytosolic fraction, as previously described. For nuclear extraction, high-salt lysis buffer (500 mM NaCl, 50 mM Tris–HCl pH 7.5, 1% (vol/vol) NP-40) supplemented with 2× protease inhibitor cocktail was added to the nuclear pellets, followed by rotation at 4°C for 2 hr. Subsequently, samples were centrifuged at 15,000 rpm at 4°C for 15 min, and the soluble portion containing the nuclear extract was combined with the cytosolic fraction. Equal volumes of clarified and normalised cytosolic and nuclear extracts were incubated with 15–20 µl GFP-Trap Agarose (ChromoTek) for overnight at 4°C with rotation. Afterwards, beads were recovered by centrifugation, washed three times in washing buffer and eluted in 35 µl of 2× SDS–DTT sample for 5 min at 95°C.

## Live cell confocal microscopy

Cells were grown on 35 mm live cells imaging dishes (MatTek), transfected as required, and maintained in culture for 48 hr after transfection. Live cell microscopy was conducted using an inverted confocal laser scanning microscope (Zeiss LSM880) in a 37°C/5% vol/vol chamber with a ×64 oil immersion objective lens with 1 Airy unit pinhole using the 488 nm (GFP; mGL), 405 nm [4',6-diamidino-2-phenylindole (DAPI)], and/or 594 nm (mCherry) lasers as appropriate. Co-localisation analysis between fluorescence channels (Pearson correlation coefficient) within individual cells was performed using Volocity software, version 6.3 (PerkinElmer).

## RNA isolation and RT

To extract total RNA, cells were treated with TRIzol Reagent (Invitrogen) for 10 min and then transferred to a new tube. Chloroform (200 µl) was added into each sample, vortexed for 1 min, and then centrifuged at 13,500 × *g*, at 4°C for 15 min. The upper clear phase was collected and mixed with equal volume of 70% ethanol. Afterward, PureLink RNA mini kit (Invitrogen) was used for washing and elution. RNA yield was assessed using NanoDrop, and 2 µg of RNA from each sample was evaluated for integrity via electrophoresis on a 1.2% agarose gel.

For RT-PCR, 2 µg of RNA was initially heated up at 70°C with RT buffer (Thermo Scientific; Cat #EP0441), 0.5 mM deoxynucleotide triphosphate (dNTPs) and 0.05 mM Oligo dT18 for 10 min. Subsequently, the reaction was supplemented with 0.5 µl RevertAid Reverse Transcriptase (Thermo Scientific) and 100 mM DTT, followed by incubation at 42°C for 90 min. The resulting cDNA product was diluted up to 1:4.

## PCR analysis of XBP1 mRNA splicing

XBP1s and XBP1u fragments were amplified from cDNA by PCR using primers flanking the splicing site identified by IRE1 with NEB Q5 High-Fidelity 2X Master Mix, as previously reported (*Neidhardt et al., 2024*). Briefly, XBP1u (255 bp) and XBP1s (229 bp) were separated in a 3% agarose gel by electrophoresis and stained with SYBR Safe nucleic acid gel stain (Invitrogen). Additionally, a XBP1[hybrid] band (280 bp) was observed. The percentage of XBP1s was quantified by determining the band intensity with Fiji, v1.53c. For quantification, it was assumed that 50% of the XBP1[hybrid] signal represented XBP1s, while the remaining 50% represented XBP1u.

## Immunoprecipitation and FURIN cleavage assay

To investigate whether ATF6α is directly cleaved by FURIN, GFP-ATF6α_LD[S1P,S2Pmut] (UK3122) construct was expressed and purified from an *FURINΔ* clone (#2.2.G). Equal volumes of clarified and normalised cell lysates were incubated with 15–20 µl GFP-Trap Agarose (ChromoTek) for overnight at 4°C with rotation. Afterwards, beads were recovered by centrifugation, washed three times in washing buffer and then incubated with 75 µl of FURIN cleavage buffer (20 mM 4-(2-hydroxyethyl)-1-piperazineethanesulfonic acid (HEPES), 1 mM CaCl₂, 0.2 mM β-mercaptoethanol, 0.1% Triton X-100; pH 7.5) supplemented with 1 U FURIN (#P8077S, NEB) for 6 hr at 25°C. As a negative control, calcium-depleted FURIN buffer was used. Additionally, an FURIN control substrate, MBP5-FN-paramyosin-ΔSal (#E8052S, NEB) containing an

FURIN site, was included in parallel reactions and incubated for 6 hr at 25°C. The eluted protein was then analysed either via immunoblot or Coomassie blue staining.

### Analysis of ATF6α redox status

Both *2K*-parental cells and *2K-CRTΔ* derivative cells were cultured in 10-cm dishes for 2 days and then treated with 2DG (4 mM) for 3 hr before harvested. Following treatment, cells were harvested in PBS containing 1 mM EDTA and 20 mM NEM and lysed in Nonidet-lysis buffer supplemented with 20 mM NEM. All subsequent buffers were also supplemented with 20 mM NEM. Equal volumes of clarified and normalised cell lysates were incubated with 15–20 µl GFP-Trap Agarose (ChromoTek) for overnight at 4°C with rotation. Afterwards, beads were recovered by centrifugation, washed three times in washing buffer/NEM and eluted in SDS/NEM loading buffer. After heating at 95°C, eluted proteins were split into two equal samples and separated under reducing (+50 mM DTT) or non-reducing SDS/ PAGE, followed by immunoblot.

### Statistics

Data groups were analysed as described in the figure legends using GraphPad Prism10 software. Differences between groups were considered statistically significant if $*p < 0.05$; $**p < 0.01$; $***p < 0.001$; and $****p < 0.0001$. All error bars represent mean ± standard error of the mean.

### Acknowledgements

We thank the CIMR flow cytometry core facility team (Reiner Schulte, Chiara Cossetti, and Gabriela Grondys-Kotarba), the microscopy team (Matthew Gratian and Mark Bowen) for technical support and the Huntington lab for access to the Octet machine. We also thank Marcella Ma (CRUK) for assistance with NGS and Avi Ashkenazi (Genentech) for the monoclonal antibody against human IRE1_LD. This work was supported by Wellcome Trust Principal Research Fellowship to DR (Wellcome 224407/Z/21Z).

## Additional information

### Competing interests

David Ron: Reviewing editor, *eLife*. The other authors declare that no competing interests exist.

### Funding

| Funder | Grant reference number | Author |
|---|---|---|
| Wellcome Trust | 10.35802/224407 | David Ron |

The funders had no role in study design, data collection, and interpretation, or the decision to submit the work for publication. For the purpose of Open Access, the authors have applied a CC BY public copyright license to any Author Accepted Manuscript version arising from this submission.

### Author contributions

Joanne Tung, Data curation, Formal analysis, Investigation, Methodology, Writing - review and editing; Lei Huang, Formal analysis, Investigation, Methodology; Ginto George, Investigation, Methodology; Heather P Harding, Resources, Investigation, Methodology, Writing - review and editing; David Ron, Conceptualization, Supervision, Funding acquisition, Investigation, Visualization, Writing - review and editing; Adriana Ordonez, Conceptualization, Data curation, Formal analysis, Supervision, Investigation, Methodology, Writing - original draft, Project administration

### Author ORCIDs

Joanne Tung ⓘ http://orcid.org/0009-0006-3733-8218
Ginto George ⓘ https://orcid.org/0000-0003-4804-9594
Heather P Harding ⓘ https://orcid.org/0000-0002-7359-7974
David Ron ⓘ https://orcid.org/0000-0002-3014-5636
Adriana Ordonez ⓘ https://orcid.org/0000-0003-4239-459X

Reviewer #1 (Public review): https://doi.org/10.7554/eLife.96979.3.sa1
Author response https://doi.org/10.7554/eLife.96979.3.sa2

## Additional files

### Supplementary files

• Supplementary file 1. Ranking of genes enriched in the *activator and repressor ATF6α screens*. Genes are ranked by 'pos | rank' value. Notably, *MBTPS1*, encoding for S1P protease, is on the top of positively selected genes in the *activator screen*, while *SETDB1* (SET Domain Bifurcated Histone Lysine Methyltransferase 1) is on the top position among the positively selected genes in the *repressor screen*. The top 100 positively selected genes correspond to *Figure 1C, D*.

• Transparent reporting form

### Data availability

All data generated or analysed during this study are included in the manuscript, supporting files, or are submitted to public databases. The raw and processed high-throughput sequencing data reported in this paper have been uploaded in NCBI's Gene Expression Omnibus (GEO, accession number: GSE254745).

The following dataset was generated:

| Author(s) | Year | Dataset title | Dataset URL | Database and Identifier |
|-----------|------|---------------|-------------|------------------------|
| Ordoñez A, Harding PH, Ron D | 2024 | A genome wide CRISPR/Cas9 screen identifies calreticulin as a selective repressor of ATF6α | https://www.ncbi.nlm.nih.gov/geo/query/acc.cgi?acc=GSE254745 | NCBI Gene Expression Omnibus, GSE254745 |

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
