## [Editor Report · eLife assessment]

In this **important** study, the authors explore ER stress signaling mediated by ATF6 using a genome-wide gene depletion screen. They find that the ER chaperone Calreticulin binds and directly represses ATF6, a new and intriguing function for Calreticulin. The evidence presented is **convincing**, based on CHO genetics and biochemical analysis.

---

## [Referee Report · Reviewer #1 (Public review)]

Summary:

In this manuscript, Tung and colleagues identify Calreticulin as a repressor of ATF6 signaling using a crispr screen and characterize the functional interaction between ATF6 and CALR.

Strengths:

The manuscript is well written and interesting with an innovative experimental design which provides some new mechanistic insight into ATF6 regulation as well as crosstalk with the IRE1 pathway. The methods used were fit for purpose and reasonable conclusions were drawn from the data presented.

Comments on latest version:

The authors did a good job at addressing my comments even though they found several aspects to exceed the scope of the work. The manuscript is clearer now and the model pushed by the authors is better supported by the data. One point I am curious about the authors' opinion would be about the status of ATF6alpha activation in pathological cells in which CALR is mutated (e.g., myeloproliferative neoplasms), although this neither challenges the conclusions of the manuscript and my positive opinion of the work.

---

## [Author Response]

The following is the authors’ response to the original reviews.

**eLife assessment**
The authors explore ER stress signalling mediated by ATF6 using a genome-wide gene depletion screen. They find that the ER chaperone Calreticulin binds and directly represses ATF6; this proposed function for Calreticulin is intriguing and constitutes an important finding. The evidence presented is based on CHO genetic evidence and biochemical results and is convincing.

We thank the editors for their favourable assessment of our work.

**Reviewer #1 (Public Review):**
Summary:In this manuscript, Tung and colleagues identify Calreticulin as a repressor of ATF6 signalling using a CRISPR screen and characterize the functional interaction between ATF6 and CALR.Strengths:The manuscript is well written and interesting with an innovative experimental design that provides some new mechanistic insight into ATF6 regulation as well as crosstalk with the IRE1 pathway. The methods used were fit for purpose and reasonable conclusions were drawn from the data presented. Findings are novel and bring together glycoprotein quality control and activation of one sensor of the UPR. This is a novel perspective on how the integration of ER homeostasis signals could be sensed in the ER.

We thank the reviewer for their favourable assessment of our work.

Weaknesses:Several points remain to be documented to support the authors' model.Major comments(1) It is interesting that BiP, PDIs, and COPII are not identified in the screen. Might this indicate some bias in the system perhaps limiting its sensitivity or pleiotropic effects of the reporter?

The reviewer raises a valid concern. Our CRISPR screen aimed to identify genes that selectively modulate ATF6⍺. Therefore, we excluded from consideration genes whose inactivation had effects on the broader ER environment. This would disfavour the selection of genes encoding BiP, PDI and COPII components. Additionally, a positive selection screen inherently removes essential genes like BiP. The absence of COPII components among the hits could be due to essentiality or that those components are not strong selective modulators for ATF6⍺ activation, as the stronger ATF6⍺ modulators as S1P, S2P and transcription factor S2P and NFY were among our top hits. Cell type specificity may also play a role. For example, ERp18, a small PDI previously implicated in ATF6⍺ activation (Oka et al 2019; PMID: 31368601), despite the presence of sgRNAs targeting hamster ERp18 in the library. Interestingly, depletion of ERp18 in our dual UPR reporter CHO-K1 cell line did not affect the ATF6⍺ and IRE1⍺ UPR branches in CHO-K1 cells. This new information has been incorporated into the revised manuscript as Supplemental Figure S6E and the discussion has been edited in line with these comments.

(2) CLR interacts with ATF6 independently of ATF6 glycans (and cysteines). How do the authors reconcile this observation with the lectin functions of CALR? What is the interaction mode then - if the CALR N (lectin) domain is not involved, is it the P domain that is responsible for the interaction? All the binding experiments are performed in the presence of 1 mM CaCl2, is calcium necessary for CALR to achieve binding?

These points merit clarification. The Biolayer Interferometry (BLI) assay reported on an interaction between ATF6 and CRT that is independently of ATF6⍺ glycans. However, cellbased experiments revealed a contribution of glycan-dependent interactions to the binding and repression. Therefore, we conclude that the interaction of CRT with ATF6⍺ likely involves both lectin-dependent and lectin-independent interactions (dependent on the P-domain). Indeed, this hybrid model has previously been suggested as the mode of stable interaction of CRT with other substrates, as cited in the discussion section (Wijeyesakere et al., 2013; PMID: 24100026). CRT is a known calcium-dependent protein, and all the in vitro experiments were conducted in the presence of 1 mM CaCl2. We do not have data from experiments without CaCl2.

(3) Does the introduction of the reporter system affect the normal BiP (or ATF6) protein levels in the cells?

To address this question, we have conducted new experiments comparing endogenous BiP protein levels between the reporter-containing cells and the parental CHO-K1 cells using immunoblotting and an anti-BiP antibody. These data indicate that the reporter system does not affect to the endogenous BiP protein levels. This new information has been incorporated as revised Supplemental Figure S1C.

(4) Does the depletion of CRT affect BiP interaction with ATF6? The absence of CRT may lead to misfolding of glycoproteins and titration of BiP away from ATF6 leading to activation. An indicator of ER stress levels that is independent of ATF6 and IRE1 might be useful.

To further assess ER stress levels in CRT-depleted cells, we compared expression levels of endogenous ER resident proteins containing a KDEL signal (e.g., P3H1, GRP94, BiP and PDI) in parental CHO-K1 cells, dual UPR reporter cell lines (XC45-6S) and CRT-depleted cells (CRT∆#2P) under basal conditions and during ER stress by immunoblotting. This comparison confirmed the basal elevation in BiP protein level in cells lacking CRT, consistent with previous findings (Figure 2D) and more broadly the integrity of UPR signalling in cells lacking CRT. In the interest of time, we did not extend the analysis to other branches of the UPR. This new information has been incorporated as Supplemental Figure S5 and in the text of the revised manuscript.

(5) Does CALR depletion alter ATF6 redox status.

We thank the reviewer for raising this interesting point. In response, we compared ATF6⍺ redox status in parental and CRT-depleted cells using non-reducing SDS-PAGE. Overall, the redox pattern was similar in parental and CRT-depleted cells with the detection of two redox forms: an inter-chain disulfide-stabilised dimer and the monomer. Under basal conditions, ATF6⍺ predominantly existed as a monomer, while under ER stress, the monomer band decreased with a corresponding increase in a disulfide-stabilised dimer form in parental cells, as previously reported (Oka et al, 2022; PMID: 35286189). However, under ER stress, CRTdepleted cells showed a significantly higher fraction of monomer versus dimer compared to parental cells. Taking all together, these data suggest that the loss of CRT may favour the monomeric form of ATF6α, which is proposed to be more efficiently trafficked (Nadanaka, et al 2007; PMID: 17101776), aligning with our observations that CRT depletion is associated to constitutive activation of ATF6α. These new data have been included as Supplemental Figure S7 and are detailed explained in the results section of the revised manuscript.

(6) Figure 4C would benefit from some immunoblotting against BiP.

Although we acknowledge the validity of this suggestion and understand the referee's interest in comparing the amount of CRT in pulldown with that of BiP, the necessity of generating additional samples makes this experiment impractical. Consequently, we opted not to include in our conclusion any comparison regarding the retention of ATF6α by BiP relative to CRT.

(7) Overlooked requirement of cysteines for ATF6 functionality (Figure 5B).

We interpret this comment to refer to the inactivity of the cysteine-free allele of ATF6⍺. Whilst this is a reproducible observation of significance to the structure-activity features of ATF6⍺’s luminal domain, it is less informative in terms of understanding trans-active regulators of ATF6⍺ and was therefore not explored further.

(8) Without a clear definition of the role of CRT in ATF6 folding, one cannot infer that the observed phenotype is not based on defects in ATF6 "folding" and glycosylation considering the possibility of activation of newly synthesised un-glycosylated ATF6.

If the main role of CRT were to assist ATF6⍺ folding, one would expect that depletion of CRT would lead to a non-functional ATF6⍺, resulting in ER retention and less activity. However, our data indicate that the loss of CRT correlates with the constitutive activation of the ATF6⍺ fluorescent reporter and increased Golgi trafficking and processing of ATF6⍺. Therefore, these data suggest that in CRT-depleted cells, the majority of ATF6⍺ is likely to fold to a functional state.

(9) ATF6 was defined in several studies as a natively unstable protein and shows a close relationship with the ERAD machinery, is the role of CALR also involved in a quality control mechanism for natively unfolded ATF6?

The reviewer brings up a valid point too. Although we have not closely evaluated the role of CRT in the quality control machinery, we observed that the loss of CRT was not associated with an increased levels of ATF6⍺ in CRT depleted cells in basal conditions compared with parental cells (Fig 3B.1, compare line 1 and line 7; Figure 3B.2, compare line 1 and line 5). These observations suggest that if ATF6⍺ were degraded by ERAD and loss of CRT compromised ERAD functionality, CRT-depleted cells should exhibit increased levels of endogenous ATF6⍺. The fact that endogenous ATF6⍺ levels are slightly reduced in CRT depleted cells does not support a role for CRT in the quality control mechanism for natively unfolded ATF6⍺.

(10) C618 in ATF6 is located within the BiP binding site and in close proximity of an Nglycosylation site. Is this region of particular importance for CALR binding?

It is an interesting point that we have not explored in this study. Consequently, without experimental data, we cannot infer the possible implications of C618 in CRT binding.

(11) The authors have mutated all the N glycosylation sites at once; they should be mutated one by one and the impact on ATF6 stability evaluated independently of the CALR status.

We agree that analysing each N-glycosylation site individually would provide further insight into their contributions to ATF6⍺ stability/functionality. However, given the scope of the paper in its present form we have elected not to addressing this point.

(12) The relationship between the absence of CALR and IRE1 remains weak. The authors do not exclude the possibility that CALR could have a direct effect on IRE1 itself. This should be either removed or further investigated.

We beg to differ. The relationship between the absence of CRT and IRE1 is not weak; loss of CRT in CHO-K1 cells represses IRE1; we conceded readily that the relationship is incompletely understood. ATF6⍺ signalling involves crosstalk with the IRE1 pathway, partly mediated by direct heterodimerisation of N-ATF6⍺ with XBP1s (Yamamoto et al., 2007, 2004). Additionally, recent research has shown that ATF6⍺ activity can repress IRE1 signalling (Walter et al., 2018). Therefore, given that our results indicate that the loss of CRT leads to constitutive activation of ATF6⍺, we suggest that a negative feedback loop in which ATF6⍺ represses IRE1 contributes to the observations made here on the relationship between CRT and IRE1. This does not exclude other aspects to the relationship, a point that is now clarified further in the revised manuscript.

Minor pointIn the introduction on page 3 it is mentioned that loss of ATF6 impairs survival in cellular and animal models, this is not completely true as ATF6a ko in mice has no clear deleterious phenotype and only the double ko ATF6a/b has some dramatic impact.

We have modified that sentence on the revised manuscript.

**Reviewer #2 (Public Review):**
Summary:In this study, the authors set out to use an unbiased CRISPR/Cas9 screen in CHO cells to identify genes encoding proteins that either increase or repress ATF6 signalling in CHO cells.Strengths:The strengths of the paper include the thoroughness of the screens, the use of a novel, double ATF6/IRE1 UPR reporter cell line, and follow-up detailed experiments on two of the findings in the screens, i.e. FURIN and CRT, to test the validity of involvement of each as direct regulators of ATF6 signalling. Additional strengths are the control experiments that validate the ATF6 specificity of the screens, as well as, for CRT, the finding of focus, determining roles for the glycosylation and cysteines in ATF6 as mechanistically involved in how CRT represses ATF6, at least in CHO cells.

We thank the reviewer for their favourable assessment of our work.

Weaknesses:(1) The weaknesses of the paper are that the authors did not describe why they focused only on the top 100 proteins in each list of ATF6 activators and repressors.

We concede that the more genes one studies the better. However, In whole genome CRISPR screens where thousands of hits arise, it is a common practise that researchers prioritise candidates with the greatest significant as those genes are likely to have a more meaningful impact on the phenotype under investigation. Therefore, our decision to focus on the top 100 genes was based on a desire to identify the most prominent and potentially impactful candidates for further analysis, ensuring a manageable scope for in-depth study while maintaining a measure of relevance and significance. Moreover, setting the threshold at 100 hits to perform GEO enrichment analysis is a practise used by previous researchers (PMID: 30323222; PMID: 37251921). In our case, the top 100 hits included the genes with an adjusted P < 0.005. For interested readers, the full ranked list is accessible in the GEO databank (GSE254745) and as supplemental Table S1.

(2) Additionally, there were a few methodology items missing, such as the nature of where the insertion site in the CHO cell genome of the XBP1::mCherry reporter. Since the authors go to great lengths to insert the other reporter for ATF6 activation in a "safe harbor" location, it leads to questions about whether the XBP1::mCherry reporter insertion is truly innocuous.

We appreciate the opportunity to clarify certain aspects of our experimental procedures. In order to generate a double UPR reporter cell line, we employed a previously established the XC45 CHO-K1 clone with an integrated XBP1s::mCherry reporter (Harding et al., 2019; PMID: 31749445). Since the ROSA26 safe harbor locus was available in the XC45 CHO-K1 cell line, we directed integrated the ATF6⍺ reporter there. To provide further clarity, the revised manuscript includes additional details in the Methods section regarding the creation of the XBP1 reporter.

(3) An additional weakness is that the evidence for the physical interaction between ATF6LD and CRT is not strong, being dependent mainly on a single IP/IB experiment in Figure 4C that comprises only 1 lane on the gel for each of the test cases. Moreover, while that figure suggests that the interaction between CRT and ATF6 is decreased by mutating out the glycosylation sites in the ATF6LD, the BLI experiment in the same figure, 4B, suggests that there are no differences in the affinities of CRT for ATF6LD WT, deltaGly and deltaCys.

We would like to highlight that in the IP/IB experiments (see Figure 4C), where wildtype ATF6 (ATF6⍺_LDWT) and GFP-ATF6_LD∆Gly were transiently transfected, GFP-ATF6_LD∆Gly was expressed at lower levels than ATF6⍺_LDWT. This lower expression levels might explain why CRT is more prominently immunoprecipitated with ATF6⍺_LDWT and could account for the differences observed among in vitro and *in vivo* assays.

(4) An additional detail is that I found Figure 6A to be difficult to interpret, and that 6B was required in order for me to best evaluate the points being made by the authors in this figure.

We have simplified Figure 6A in the revised manuscript to make it more interpretable by focussing the reader’s attention on the transfected population.

Overall, I believe that this work will positively impact the field as it provides a list of potential regulators of ATF6 activation and repression that others will be able to use as a launch point for discovering such interactions in cells and tissues or interest beyond CHO cells. However, I agree with the authors that these findings were in CHO cell lines and that it is possible, if not likely, that some of the interactions they found will be cell type/line specific.

We accept this point and re-emphasize the qualification that our conclusions cannot be glibly extrapolated to other cell lines.